# Carbon storage in Sichuan Province (Southwest China) from 1980 to 2050: Spatial-temporal variation, driving factors and future trends

Qinglian Deng[1], Yuqi Guan[1], Xiong Duan [1]*, Bin Chen[2], Kun Zeng[1]

1 School of Geographical Sciences, Sichuan Provincial Engineering Research Center of Monitoring and Control for Soil Erosion in Dry Valleys, China West Normal University, Nanchong, China, 2 School of Geography and Environment, Liaocheng University, Liaochen, China

* duanxiong00@163.com

## Abstract

Research on carbon storage is crucial for guiding regional sustainable development. However, Sichuan Province lacks long-term systematic analyses of carbon storage, and the driving mechanisms behind its changes remain unclear. This study systematically examines the spatiotemporal evolution of LUCC(land use/cover change) and carbon storage in Sichuan from 1980 to 2020, analyzes driving factors of carbon storage changes, and simulates future carbon storage distribution under different scenarios, based on LUCC data and 13 driving factors. Key findings include: (1) Over the 40-year period, land use was dominated by grassland, forest land, and farmland, maintaining a stable "grassland/forest land in the west, farmland in the east" pattern, with notable farmland and water body shrinkage alongside grassland and construction land expansion. (2) Total carbon storage showed minor fluctuations (9,201.53–9,209.52 Tg) but exhibited significant spatial heterogeneity, persistently displaying a "high in the west and low in the east" distribution. Water body-to-grassland and farmland-to-forest land conversions substantially increased carbon storage, while forest land-to-grassland and farmland-to-construction land transitions decreased it. (3) Spatial autocorrelation analysis revealed a negative correlation between carbon storage and land use intensity, with pronounced spatial clustering—High-High clusters concentrated in western regions and Low-Low clusters distributed peripherally. (4) Temperature and Digital Elevation Model emerged as dominant factors, while transportation accessibility and precipitation showed minimal influence. Human activities demonstrated moderate regulatory effects, with factor interactions significantly enhancing explanatory power, indicating multi-factor driven changes. (5) Multi-scenario projections (2030–2050) maintained the "high in the west and low in the east" pattern. Compared to 2020, SSP1–1.9 (Shared Socioeconomic Pathway 1–1.9) showed minimal change (10,711.94~10,712.16 Tg), SSP2–4.5 (Shared Socioeconomic Pathway 2–4.5) exhibited the largest decline (9,243.73~9,202.01

**Data availability statement:** All processed datasets used in this study have been archived in our GitHub repository (https://github.com/931547291-max/Qinglian-Deng), ensuring that both the raw and derived data required to reproduce the results are fully accessible and traceable.

**Funding:** This research was supported by the National Natural Science Foundation of China (NSFC) (Grant No. 42201006), the Natural Science Foundation of Sichuan Province (Grant No. 2022NSFSC1177), and the Fundamental Research Funds of China West Normal University (Project No. 25kx005).

**Competing interests:** The authors have declared that no competing interests exist.

**Abbreviations: LUC**C: Land use/cover change; **IPCC**: Intergovernmental Panel on Climate Change; **InVE**ST: Integrated Valuation of Ecosystem Services and Trade-offs; **PLU**S: Patch-generating Land Use Simulation; **FLU**S: Future Land Use Simulation; **CMI**P6: Coupled Model Intercomparison Project Phase 6; **SSP**: Shared Socioeconomic Pathway; **DEM**: Digital Elevation Model; **NDV**I: Normalized Difference Vegetation Index; **NPP**: Net Primary Productivity; **GDP**: Gross Domestic Product; **SHA**P: SHapley Additive exPlanations; **LUDI**: Land Use Degree Index; **MAE**: Mean Absolute Error; **CA**: Cellular Automaton; **LISA**: Local Indicators of Spatial Association; **ML**: Machine learning; **LR**: Logistic Regression; **RF**: Random Forest; **SVM**: Support Vector Machine; **KNN**: K-Nearest Neighbor.

Tg), and SSP5–8.5 (Shared Socioeconomic Pathway 5–8.5) also decreased notably (9,015.01∼8,980.07 Tg). This study provides a scientific basis for future land use optimization and carbon sink management in Sichuan Province.

## 1. Introduction

The terrestrial ecosystem carbon pool is a vital component of the global carbon pool, characterized by significant interannual fluctuations. This variability allows terrestrial ecosystems to partially mitigate or amplify a substantial proportion of anthropogenic fossil fuel emissions, which is why they are referred to as a "climate regulation valve." This pool plays a crucial role in addressing climate change and regulating the global carbon cycle [1–4]. According to the Sixth Assessment Report by the IPCC (Intergovernmental Panel on Climate Change), the Earth's average temperature has increased by 1.1°C compared to pre-industrial levels, with human activities causing unprecedented climate warming over the past 2,000 years and leading to a simultaneous rise in the frequency and intensity of extreme weather events. As a result, accurately quantifying the spatiotemporal evolution of regional carbon storage and predicting its future trends has become a central scientific challenge for nations aiming to fulfill their commitments under the Paris Agreement to achieve "carbon neutrality." As the world's largest carbon emitter, China made a solemn commitment in 2020 to peak carbon emissions by 2030 and achieve carbon neutrality by 2060, incorporating the goal of "enhancing ecosystem carbon sink capacity" into the National Strategy for Climate Change Adaptation 2035. Sichuan Province, located at the core of the ecological barrier on the upper Yangtze River, serves as a transitional zone between the Tibetan Plateau and the eastern monsoon region, featuring a complex topography that includes " alpine, sub-alpine, hilly, and plain" levels, making it one of the most topographically complex and climatically sensitive regions globally [5]. The ongoing transformation of the energy structure, rapid urban expansion, and increasing occurrences of extreme weather events are reshaping Sichuan's carbon balance at an unprecedented pace [6,7]. Therefore, clarifying the patterns of spatiotemporal evolution of carbon storage in this region over the past 40 years and systematically forecasting multiple scenarios for the next 30 years has become a key scientific question for validating whether China's dual carbon strategy—aiming for peak carbon emissions by 2030 and carbon neutrality by 2060—can be effectively implemented in a complex topography-socioeconomic coupling zone.

Current research on terrestrial ecosystem carbon storage primarily utilizes three methods: field measurements, remote sensing inversion, and model simulations [8]. Field measurements involve collecting carbon data through field sampling and aggregating it via interpolation to estimate regional carbon storage. While this method can accurately derive carbon density data, it consumes a lot of time and resources. For instance, Zeng and Sun enhanced the accuracy of carbon storage estimates in the Yellow River Basin by employing Kriging interpolation on a large dataset of carbon density sampling points, creating a spatial distribution dataset for carbon density [9].

Conversely, remote sensing inversion leverages remote sensing data along with relevant models or algorithms to monitor and analyze surface and vegetation cover, allowing for the estimation of carbon storage in specific areas with advantages of efficiency and non-destructiveness [4,8]. For example, Illarionova et al. advanced forest carbon storage mapping using machine learning and stratified satellite imagery [10], while Cheng et al. employed Landsat 8 multispectral imagery and the Google Earth Engine platform to estimate forest carbon storage in Yunnan Province, China [11].

Empirical models such as the InVEST (Integrated Valuation of Ecosystem Services and Trade-offs) model, co-developed by Stanford University and the World Wide Fund for Nature, provide a simple and efficient approach that overcomes the complexities, high costs, and poor visibility associated with traditional models, making them particularly effective in assessing the impacts of climate and land use changes on ecosystem carbon storage [12]. Numerous scholars have utilized the InVEST model for carbon storage estimation. For example, Piyathilake et al. assessed carbon storage in Uva Province, Sri Lanka, using the InVEST model, revealing significant differences in carbon storage across various land use types [13]. Similarly, Ismaili Alaoui et al. estimated carbon storage in the Beht Basin of Morocco using the InVEST model, examining the relationship between land use types and carbon storage; their findings indicated that construction land and farmlands increased by 136% and 86%, respectively, with these changes exerting the most substantial impact on carbon storage [14]. Li et al. simulated carbon storage across different land use types in Heilongjiang, China, using the InVEST model, finding a strong correlation between carbon storage and LUCC (land use/cover change), with permafrost degradation identified as a key factor influencing carbon storage variations in the region [15]. These studies not only validate the effectiveness of the InVEST model in estimating carbon storage but also provide critical data support and methodological references for regional carbon cycle research.

Moreover, recent advancements in LUCC prediction models, such as the FLUS (Future Land Use Simulation) model [16] and the PLUS (Patch-generating Land Use Simulation) model [17], are frequently applied alongside the InVEST model to simulate future spatial distributions of carbon storage. For instance, Li et al. utilized the FLUS model to simulate land use patterns in Changchun, China, under three scenarios until 2030, and employed the InVEST model to assess carbon storage from 2010 to 2030; their results indicated that the transition of farmland to construction land was the primary land use change over the past decade, resulting in significant carbon losses, with further declines anticipated without human intervention [18]. Similarly, Sun et al. coupled the PLUS and InVEST models to analyze land use and carbon storage changes in Nanjing, China, from 2000 to 2020, predicting carbon storage distribution in 2040 under various development scenarios; they found that land use changes directly impacted carbon storage, with reductions expected across all scenarios by 2040, while farmland conservation and ecological protection policies could effectively mitigate these losses [19]. Additionally, Li et al. employed InVEST and PLUS models to map ecosystem carbon storage distribution patterns in Liaoning, China, over the past 20 years, forecasting spatiotemporal characteristics of carbon storage under multiple development scenarios from 2020 to 2050; their findings confirmed that LUCC directly influences carbon storage, highlighting that increases in ecological land area can significantly enhance carbon sequestration capacity [20]. These studies not only enrich the methodological framework for carbon storage estimation but also provide vital scientific evidence for global climate change mitigation and ecosystem management.

Sichuan, as a representative region of complex terrain in China and across the globe, provides an essential case for advancing carbon-cycle research, particularly for understanding the mechanisms driving the spatiotemporal dynamics of carbon storage and projecting future trajectories. These issues not only align with the frontier topics of international carbon-cycle science but also respond to the urgent demands associated with implementing China's "dual-carbon" strategy. However, most existing studies have examined only specific subregions of Sichuan—such as the western Sichuan Plateau [21,22], the Chengdu–Deyang–Mianyang urban belt [23], and the Jialing River Basin [24] —while even province-wide assessments still suffer from limitations including short temporal coverage and relatively narrow analytical perspectives [25]. To address these shortcomings, this study introduces several advances in data, modeling, and methodological frameworks: we construct a province-wide LUCC–carbon storage evolution dataset spanning 40 years

(1980–2020) based on long-term land-use data and associated driving factors; we integrate the InVEST model with a Markov chain–driven FLUS/PLUS LUCC simulation framework and incorporate CMIP6 (Coupled Model Intercomparison Project Phase 6) scenarios (SSP1–1.9, SSP2–4.5, SSP5–8.5) to build a continuous "historical–current–future" modeling system; and we further employ the Geographical Detector together with machine-learning-based SHAP (SHapley Additive exPlanations) analysis to identify the natural and socio-economic drivers of carbon-storage change across multiple dimensions, thereby enhancing both explanatory accuracy and interpretability. The study systematically addresses the following scientific questions: What are the spatiotemporal dynamic characteristics of LUCC and carbon storage in Sichuan over the past 40 years? What are the driving mechanisms behind changes in carbon storage? How will the spatiotemporal patterns of carbon storage in Sichuan evolve from 2020 to 2050 under different climate scenarios? The results of this research aim to provide scientific support for regional land use optimization, ecological protection and restoration, and the achievement of dual carbon strategy, while also offering a replicable methodological paradigm for carbon management in globally complex topographic regions.

## 2. Overview of the study area

Sichuan Province is located in southwestern China, spanning from 97°21′E to 108°33′E and from 26°03′N to 34°19′N, with a total area of approximately 486,000 km² (Fig 1). It constitutes a vital ecological transition zone connecting the Qinghai-Tibet Plateau and the Middle-Lower Yangtze Plain, characterized by complex and diverse geomorphology with prominent stepped topography descending from western highlands to eastern lowlands. As a critical water conservation area in the upper Yangtze basin, the region exhibits distinct vertical climatic zonation ranging from subtropical humid to alpine climates, fostering diverse ecological landscapes.

## 3. Materials and methods

The datasets employed in this study primarily comprise land use data and driving factors associated with land use change (Table 1). The land use data were obtained from the Resource and Environmental Science Data Platform (http://www.resdc.cn, accessed on 25 June 2025), covering the years 1980, 1990, 2000, 2010, and 2020. After preprocessing, the original data were classified into six categories: farmland, forest land, grassland, water body, unused land, and construction land. Based on the principles of data availability, significance, and temporal relevance, a total of 13 driving factors were selected to characterize the determinants of land use change. The natural environment dataset includes the following: DEM (Digital Elevation Model) data obtained from the Geospatial Data Cloud (https://www.gscloud.cn/, accessed on 25 June 2025), with slope data derived from the DEM; mean annual temperature, mean annual precipitation, NDVI (Normalized Difference Vegetation Index), and soil type data sourced from the Resource and Environmental Science Data Platform (https://www.resdc.cn, accessed on 25 June 2025); river data acquired from the National Catalogue Service for Geographic Information (https://www.webmap.cn, accessed on 25 June 2025), with the distance to rivers calculated using Euclidean distance. The socio-economic dataset comprises NPP (Net Primary Productivity), GDP (Gross Domestic Product), and population data from the Resource and Environmental Science Data Platform (https://www.resdc.cn, accessed on 25 June 2025); railway, highway, and national highway data obtained from the National Catalogue Service for Geographic Information (https://www.webmap.cn, accessed on 25 June 2025), with distance to railways, highways, and national highways also computed using Euclidean distance. All datasets were resampled to a spatial resolution of 1 km.

In addition, the calculation of carbon storage requires carbon density data, and the correction of carbon density values necessitates the use of mean annual temperature, mean annual precipitation data. The carbon density data were primarily derived from previous research findings [21,23,26–28]. Current mean annual temperature, mean annual precipitation data were obtained from the Sichuan Statistical Yearbook (http://tjj.sc.gov.cn/scstjj/, accessed on 25 June 2025), while future mean annual temperature, mean annual precipitation data were sourced from the National Tibetan Plateau Data Center (https://data.tpdc.ac.cn/, accessed on 25 June 2025).

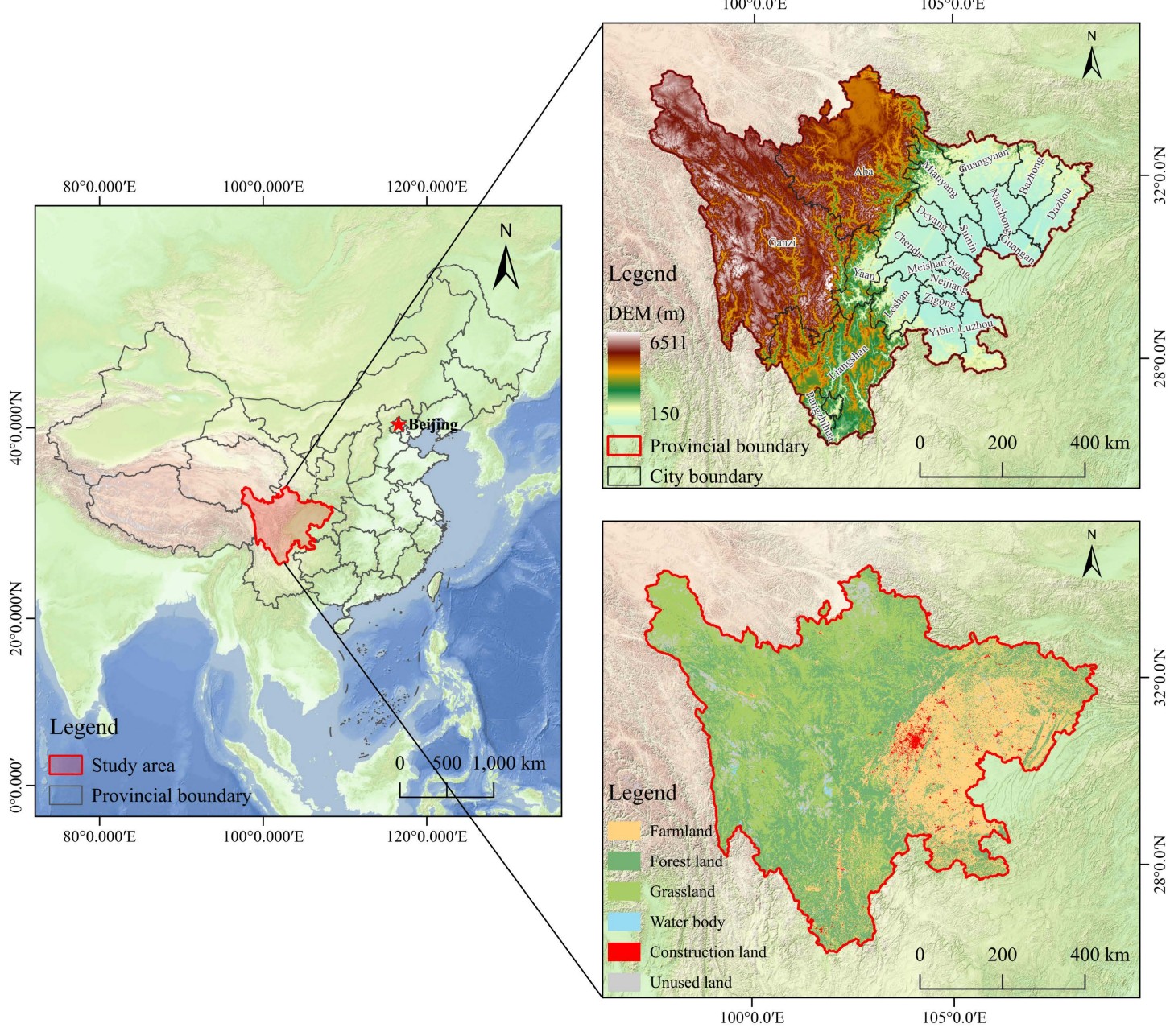

**Fig 1. Location map of the study area.**

The research framework includes three major components (Fig 2):

(1) Analyzing the spatiotemporal evolution of land use and carbon storage in the study area from 1980 to 2020;

(2) Quantifying the impacts of land-use change and other driving factors on carbon storage using spatial autocorrelation, the Geographical Detector, machine learning, and SHAP;

**Table 1. Carbon density values of different land use types in Sichuan Province (t·hm⁻²).**

| Year/ | Land use type | Farm land | Forest land | Grassland | Water body | Construction land | Unused land |
|---|---|---|---|---|---|---|---|
| **1980-2020** | $D_{above}$ | 24.85 | 39.18 | 16.05 | 12.70 | 4.40 | 12.75 |
| | $D_{below}$ | 41.45 | 78.44 | 26.55 | 37.30 | 43.65 | 68.45 |
| | $D_{soil}$ | 66.10 | 144.98 | 113.75 | 87.30 | 80.40 | 94.65 |
| | $D_{dead}$ | 1.00 | 3.50 | 1.00 | 1.00 | 0.00 | 0.00 |
| **2030-2050 SSP1–1.9** | $D_{above}$ | 33.85 | 53.37 | 21.86 | 17.30 | 5.99 | 17.37 |
| | $D_{below}$ | 56.46 | 106.85 | 36.17 | 50.81 | 59.46 | 93.24 |
| | $D_{soil}$ | 67.87 | 148.86 | 116.80 | 89.64 | 82.56 | 97.19 |
| | $D_{dead}$ | 1.36 | 4.77 | 1.36 | 1.36 | 0.00 | 0.00 |
| **2030-2050 SSP2–4.5** | $D_{above}$ | 28.16 | 44.40 | 18.19 | 14.39 | 4.99 | 14.45 |
| | $D_{below}$ | 46.98 | 88.89 | 30.09 | 42.27 | 49.47 | 77.58 |
| | $D_{soil}$ | 66.64 | 146.17 | 114.68 | 88.02 | 81.06 | 95.43 |
| | $D_{dead}$ | 1.13 | 3.97 | 1.13 | 1.13 | 0.00 | 0.00 |
| **2030-2050 SSP5–8.5** | $D_{above}$ | 25.00 | 39.42 | 16.15 | 12.78 | 4.43 | 12.83 |
| | $D_{below}$ | 41.71 | 78.92 | 26.71 | 37.53 | 43.92 | 68.87 |
| | $D_{soil}$ | 65.85 | 144.42 | 113.31 | 86.97 | 80.09 | 94.29 |
| | $D_{dead}$ | 1.01 | 3.52 | 1.01 | 1.01 | 0.00 | 0.00 |

**Note:** $D_{above}$ (aboveground carbon density), $D_{below}$ (belowground carbon density), $D_{soil}$ (soil carbon density), $D_{dead}$ (dead organic carbon density).

(3) Simulating the spatiotemporal distribution of carbon storage in Sichuan from 2030 to 2050 under multiple scenarios using the PLUS and InVEST models.

### 3.1 Analysis on the spatiotemporal changes of LUCC

The spatiotemporal dynamics of LUCC were quantified using a land-use transition matrix, in which rows and columns represent the initial and final land-use categories, and each cell records the area or proportion converted from one category to another. This matrix effectively characterizes both the direction and magnitude of land-use transitions. In addition, the Land Use Degree Index was calculated to assess the overall intensity of human activities on land use. The calculation of the Land Use Degree Index is defined as follows:

$$L = 100 \times \sum_{i=1}^{n} (D_i \times S_i)$$

(1)

In the formula, $L$ represents the Land Use Degree Index; $D_i$ denotes the intensity grading index assigned to land use type $i$ (with construction land assigned a value of 4, unused land a value of 1, farmland a value of 3, and all other land types assigned a value of 2 [29]); $n$ is the total number of land use types; and $S_i$ represents the percentage of the total area occupied by land use type $i$.

### 3.2 Prediction and future scenario setting of LUCC

LUCC prediction models can be categorized into quantity models and spatial models. By integrating these two types of models and incorporating multiple future scenarios, more accurate forecasts of future LUCC can be achieved.

**3.2.1 Quantitative prediction of LUCC.** The Markov chain model and linear regression model are two widely used approaches for forecasting future quantities of different land-use categories. The Markov chain model, rooted in stochastic process theory, estimates transition probabilities among land-use types based on historical observations and constructs a

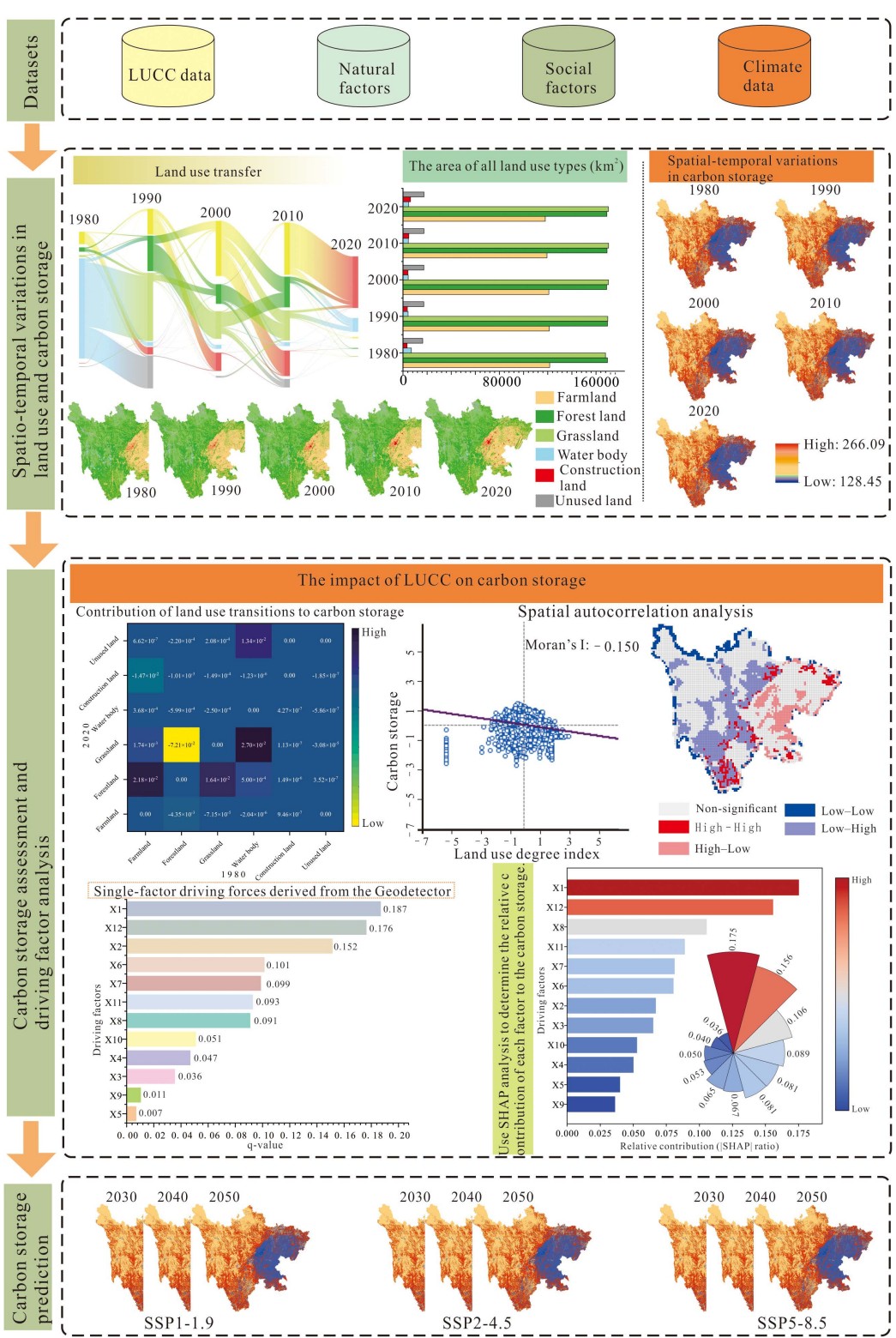

**Fig 2. Research method framework.**

transition probability matrix to project land-use quantities at future time points. In contrast, linear regression is a classical statistical technique that establishes a linear functional relationship between dependent and explanatory variables. Considering the distinct characteristics and underlying assumptions of these two methods, this study applies both models independently to forecast land-use quantities. Their predictive performance is then evaluated by comparing the MAE (Mean Absolute Error) of the simulation results, enabling the identification of the more appropriate model for LUCC quantity prediction.

**3.2.2 Spatial prediction of LUCC.** The spatial distribution of land-use change was simulated using the FLUS and PLUS models, both of which are based on the CA (Cellular Automaton) framework. The FLUS model incorporates an adaptive inertia and competition mechanism as well as a multi-type land coordination strategy, and employs an ANN (Artificial Neural Network) to capture the nonlinear relationships between terrain, climate, socioeconomic factors, and land-use categories, thereby generating cell-level suitability probabilities for simulating spatial land-use evolution [16]. Building upon this framework, the PLUS model introduces a patch-generation and patch-expansion mechanism that more realistically represents land-use transitions with pronounced patch characteristics—such as urban expansion and farmland encroachment—and provides a finer depiction of spatial competition and coordination among different land-use types [17]. In this study, multiple simulations were conducted separately using FLUS and PLUS, and the model with the higher average performance (e.g., Kappa coefficient) across repeated runs was selected. This ensemble-based approach helps mitigate the randomness inherent in single simulations and enhances the overall reliability of the spatial prediction results.

**3.2.3 Future scenario setting.** The issue of carbon cycling in terrestrial ecosystems under climate warming has attracted considerable attention. Over recent decades, various ecological – climate models have been widely employed to investigate changes in carbon storage of terrestrial ecosystems under future warming scenarios. Accordingly, this study incorporates three future scenarios (2030–2050) from CMIP6 for projecting carbon storage dynamics.

1. SSP1–1.9 (Low Emission Scenario): Under this scenario, land use transitions are subject to certain ecological constraints, aiming to maintain ecosystem stability while balancing development needs and ecological protection to promote sustainable land resource utilization. The low emission scenario emphasizes an ecologically rational, efficient, and regulated land allocation strategy that ensures development goals are met alongside ecological conservation, allowing different land use types to adjust their management practices within reasonable limits based on their inherent characteristics and ecological requirements.

2. SSP2–4.5 (Medium Emission Scenario): This scenario prioritizes strict protection of farmland, while allowing moderate conversion among forest land, grassland, water body, and unused land to support farmland supplementation, ecological restoration, and economic development. It aims to maintain stable farmland area while providing space for rational land development, reinforcing farmland protection through land use transition rules to balance ecological conservation with development demands.

3. SSP5–8.5 (High Emission Scenario): This scenario emphasizes rapid economic growth accompanied by high energy demand and greenhouse gas emissions, prioritizing economic development. Land use transitions are primarily driven by economic objectives, with ecological constraints subordinated to development needs. Flexible land conversion under this scenario facilitates accelerated economic expansion, reflecting a land use strategy that places "economic priority" at the forefront.

## 3.3 Carbon storage calculation based on the InVEST model

Carbon storage was assessed using the carbon module of the InVEST model, which partitions ecosystem carbon stocks into four components: aboveground biomass carbon, belowground biomass carbon, soil carbon, and dead organic carbon

[12]. Based on land-use categories and their corresponding carbon density values, the total carbon storage of the study area was calculated using the following equation:

$$C = \sum_{1}^{i} (D_{above-i} + D_{below-i} + D_{soil-i} + D_{dead-i}) \times A_i \tag{2}$$

In the formula, $i$ denotes the number of land use types; $A_i$ represents the area of the ith land use type; $D_{above-i}$, $D_{below-i}$, $D_{soil-i}$, and $D_{dead-i}$ correspond to the carbon densities of aboveground biomass, belowground biomass, soil organic carbon, and dead organic matter carbon for the ith land use type, respectively; and C represents the total ecosystem carbon storage.

The accuracy of carbon storage assessment in the InVEST model largely depends on the reliability of carbon density data for each land use type. Previous studies have confirmed that carbon density is primarily influenced by regional mean annual precipitation and mean annual temperature [30,31]. Therefore, based on empirical data from prior research [21,23,26–28]and carbon density correction methods [30–32], this study applies adjustments to carbon density values under the three future scenarios (Table 1). The carbon density correction formula is as follows:

$$C_{BT} = 28 \times T + 398 \tag{3}$$

$$C_{BP} = 6.7981e^{0.00541P} \tag{4}$$

$$C_{SP} = 3.3968P + 3996.1 \tag{5}$$

$$K_{BP} = (C'_{BP})/(C''_{BP}) \tag{6}$$

$$K_{BT} = (C'_{BT})/(C''_{BT}) \tag{7}$$

$$K_B = K_{BT} \times K_{BP} \tag{8}$$

$$K_S = (C'_{SP})/(C''_{SP}) \tag{9}$$

In the formula, $T$ represents the mean annual temperature, and $P$ denotes the mean annual precipitation. $C_{BT}$ and $C_{BP}$ correspond to the biomass carbon density corrected based on mean annual temperature, mean annual precipitation, respectively, while $C_{SP}$ is the soil carbon density corrected according to precipitation. $K_{BP}$ and $K_{BT}$ are the correction coefficients for biomass carbon density influenced by temperature and precipitation, respectively. $C'_{BP}$ and $C''_{BP}$ represent the precipitation-corrected biomass carbon density for the current and future scenarios in Sichuan Province, respectively; similarly, $C'_{BT}$ and $C''_{BT}$ denote the temperature-corrected biomass carbon density for current and future scenarios. $C'_{SP}$ and $C''_{SP}$ correspond to the temperature-corrected soil carbon density values. Finally, $K_B$ and $K_S$ are the correction coefficients for biomass and soil carbon density under current conditions and various future scenarios in Sichuan Province.

### 3.4 Driving factors of carbon storage

**3.4.1 Impact of LUCC on carbon storage.** Changes in land use types can significantly impact carbon storage, with varying effects depending on the specific transitions between different land use categories. This study employs a contribution rate quantification formula to systematically assess the contribution of land use changes to carbon storage variations in Sichuan Province from 1980 to 2020. The contribution rate is calculated using the following formula:

$$R = ((C_b - C_a)/C) \times (L/S) \tag{10}$$

In the formula, $C$ represents the contribution rate of carbon storage change caused by a specific land use transition to the total carbon storage change in the study area; $C_a$ and $C_b$ denote the carbon storage before and after the land use change, respectively; $C$ is the sum of the absolute values of all carbon storage changes within the region; $L$ refers to the area of land undergoing the specific type of transition; and $S$ represents the total area of the study region.

**3.4.2 Relationship between the LUDI and carbon storage.** This study utilizes grid-based LUDI (Land Use Dynamic Index) and carbon storage data at a 10 km resolution as input for spatial analysis. Using the bivariate spatial autocorrelation module in GeoDa software (version 1.22), global Moran's I and LISA (Local Indicators of Spatial Association) cluster maps were generated to investigate the spatial correlation between land use intensity and carbon storage across the study area.

**3.4.3 GeoDetector-based analysis.** GeoDetector is a nonlinear statistical framework capable of quantifying spatial stratified heterogeneity and its underlying drivers without requiring linear assumptions [33], and it has been widely applied to identify driving forces of carbon storage dynamics [34,35]. In this study, factor detection and interaction detection were performed using the q-statistic as the core indicator (with larger q-values indicating higher explanatory power) to quantify the contribution of individual factors to the spatial differentiation of carbon storage, while also assessing how explanatory power changes under factor interactions. A set of natural and anthropogenic variables was selected, including mean annual temperature (X1), slope (X2), distance to rivers (X3), distance to railways (X4), annual precipitation (X5), population (X6), NPP (X7), NDVI (X8), distance to national highways (X9), distance to expressways (X10), GDP (X11), and DEM (X12). These variables represent multiple dimensions of terrain, climate, vegetation, and human activities and exhibit strong representativeness and data availability. Soil type was excluded due to its discrete categorical characteristics, which are unsuitable for GeoDetector analysis.

**3.4.4 Machine learning and SHAP analysis.** ML (Machine learning) models do not require predefined functional forms and are capable of capturing high-order nonlinear relationships between carbon storage and multiple driving factors; however, their "black-box" nature limits interpretability. To enhance model explainability, SHAP was employed to decompose the marginal contribution of each feature based on Shapley values [36]. Seven ML algorithms—Logistic Regression, RF (Random Forest), SVM (Support Vector Machine), KNN (K-Nearest Neighbor), XGBoost, CatBoost, and LightGBM—were used to model the relationships between carbon storage and the 13 previously mentioned influencing factors. Model accuracy and the Kappa coefficient were used as evaluation metrics. The best-performing model was then selected for SHAP analysis to determine the marginal contributions of individual factors to carbon storage changes.

## 4. Results

### 4.1 Spatiotemporal changes of land use in Sichuan Province from 1980 to 2020

Overall, from 1980 to 2020, land use in Sichuan Province was dominated by grassland, forest land, and farmland, collectively accounting for over 94% of the total area. Grassland and forest land were primarily distributed in the western region of Sichuan, while farmland was concentrated in the eastern part of the study area. Construction land expanded progressively year by year, water body exhibited a noticeable decline in area, and unused land appeared sporadically across the region (Fig 3).

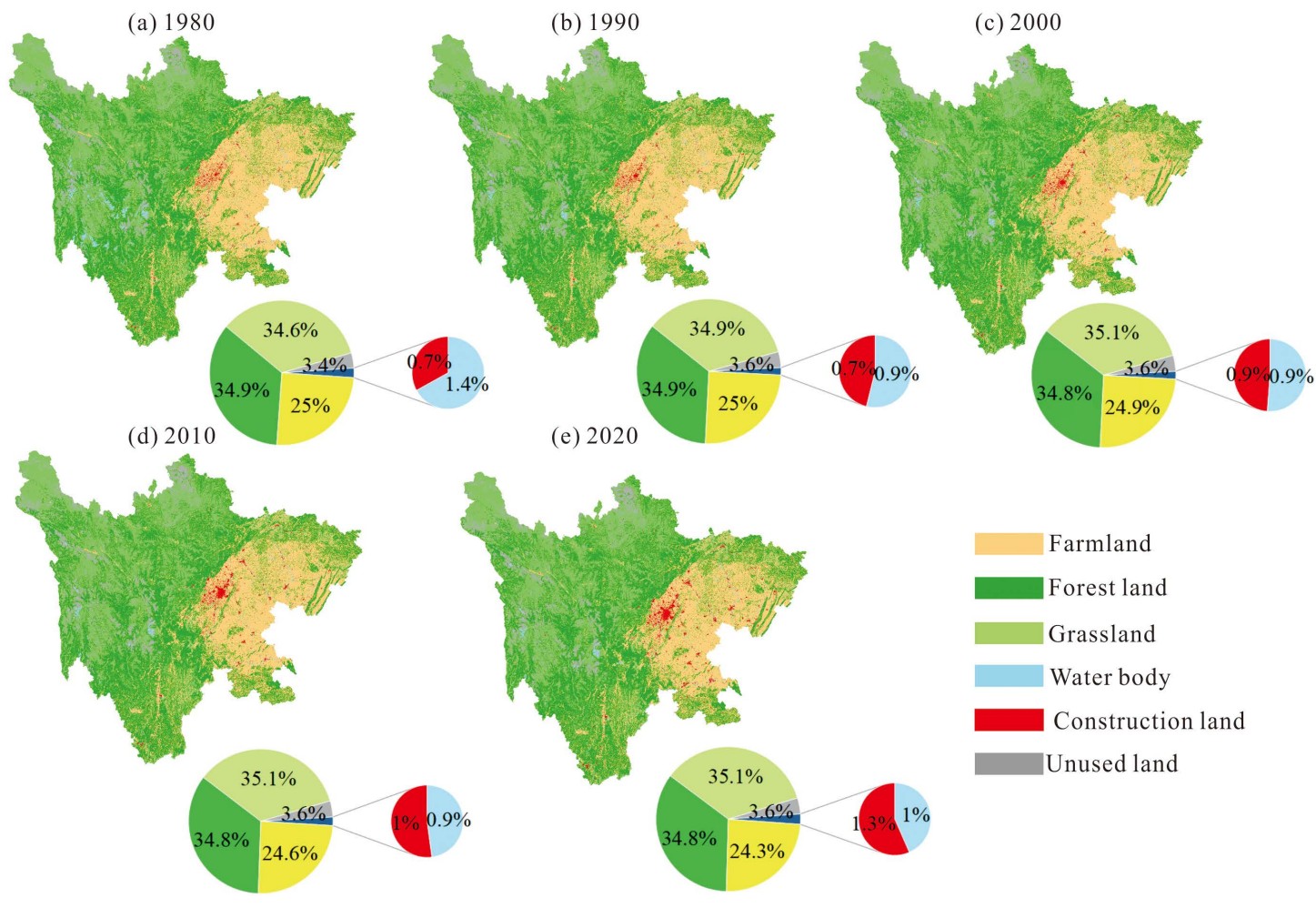

**Fig 3. Spatial distribution of land use types in Sichuan Province from 1980 to 2020.**

**Table 2. Changes in land use area in Sichuan Province from 1980 to 2020 (km²).**

| Year | Land use type | | | | | | Total |
|------|---------------|-------------|-----------|------------|-------------------|-------------|---------|
| | Farmland | Forest land | Grassland | Water body | Construction land | Unused land | |
| **1980** | 121,602 | 169,508 | 167,862 | 6,966 | 3,438 | 16,375 | 485,751 |
| **1990** | 121,317 | 169,526 | 169,724 | 4,257 | 3,640 | 17,287 | 485,751 |
| **2000** | 120,886 | 168,850 | 170,258 | 4,345 | 4,145 | 17,267 | 485,751 |
| **2010** | 119,381 | 169,162 | 170,464 | 4,441 | 4,841 | 17,462 | 485,751 |
| **2020** | 117,947 | 168,995 | 170,263 | 4,817 | 6,277 | 17,452 | 485,751 |

From 1980 to 2020, cropland experienced the most pronounced decline, decreasing from 121,602 km² to 117,947 km² (−3.0%), with most of the lost area converted to construction land (Table 2, Fig 4). Forestland remained generally stable, showing a slight decrease from 169,508 km² to 168,995 km² (−0.3%), primarily due to its conversion to grassland (Fig 4). Grassland expanded from 167,862 km² to 170,263 km² (+1.4%), mainly driven by inflows from water bodies and

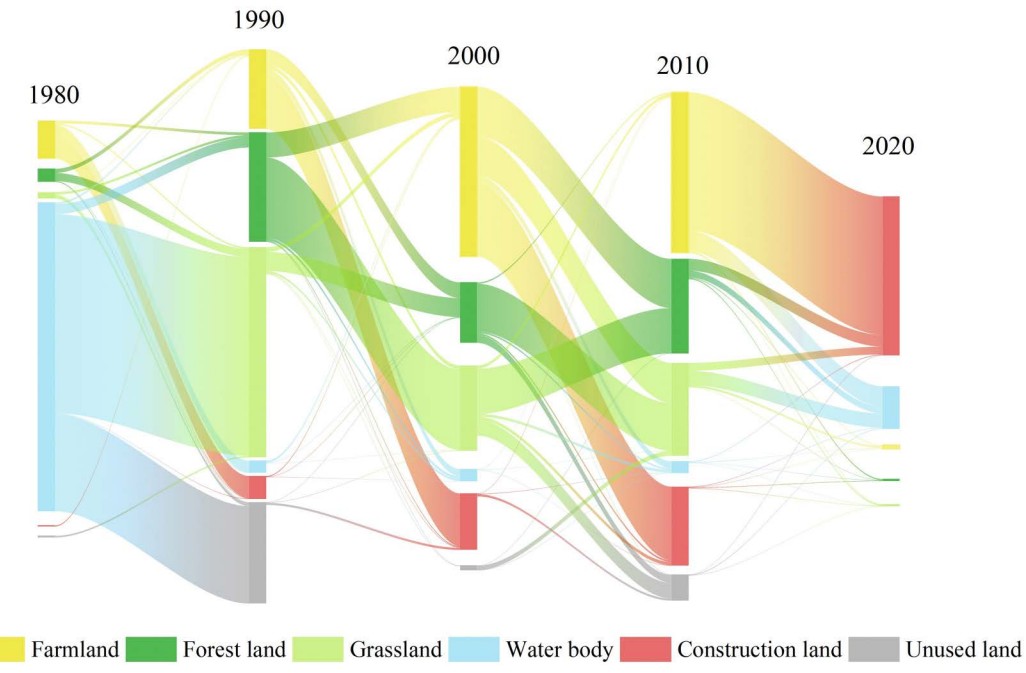

| | Farmland | | Forest land | | Grassland | | Water body | | Construction land | | Unused land |

**Fig 4. Sankey diagram of land use transitions in Sichuan Province from 1980 to 2020.**

forestland (Fig 4). Water bodies exhibited the most substantial proportional reduction, shrinking from 6,966 km² to 4,817 km² (–30.8%), largely replaced by grassland (Fig 4). Construction land showed the most dramatic expansion, increasing from 3,438 km² to 6,227 km² (+82.6%), predominantly at the expense of cropland, followed by forestland and grassland (Fig 4). Unused land increased from 16,375 km² to 17,452 km² (+6.6%), mainly due to conversions from water bodies and, to a lesser extent, grassland (Fig 4).

### 4.2 Spatiotemporal changes of carbon storage in Sichuan Province from 1980 to 2020

From a spatial perspective, carbon storage in Sichuan Province consistently exhibited a "high in the west and low in the east" pattern between 1980 and 2020. The southwestern and western regions of the study area, dominated by concentrated forest land and grassland, served as high-value carbon storage zones, whereas the central and eastern regions, characterized by higher proportions of farmland and construction land, displayed lower carbon storage levels. This spatial distribution pattern remained stable throughout the period (Fig 5). Temporally, carbon storage maintained a relatively steady state over the 40 years, with the largest increase occurring between 1980 and 1990 (7.15 Tg) and a notable decrease between 1990 and 2000 (7.99 Tg) (Table 3).

Fig 6 illustrates the spatial changes in carbon storage from 1980 to 2020. Between 1980 and 1990 (Fig 6a), several areas of significant carbon storage increase (highlighted in red) were observed, primarily concentrated in the southwestern part of the study region, while the majority of the area remained largely unchanged (shown in gray). During the period from 1990 to 2020 (Fig 6b–6d), the spatial distribution of carbon storage remained stable without notable changes. Over the entire 1980–2020 period (Fig 6e), despite isolated pockets of significant carbon storage growth, most areas exhibited minimal change, with no extensive zones of either substantial carbon loss or gain, underscoring the relative stability of carbon storage in the study area over these four decades.

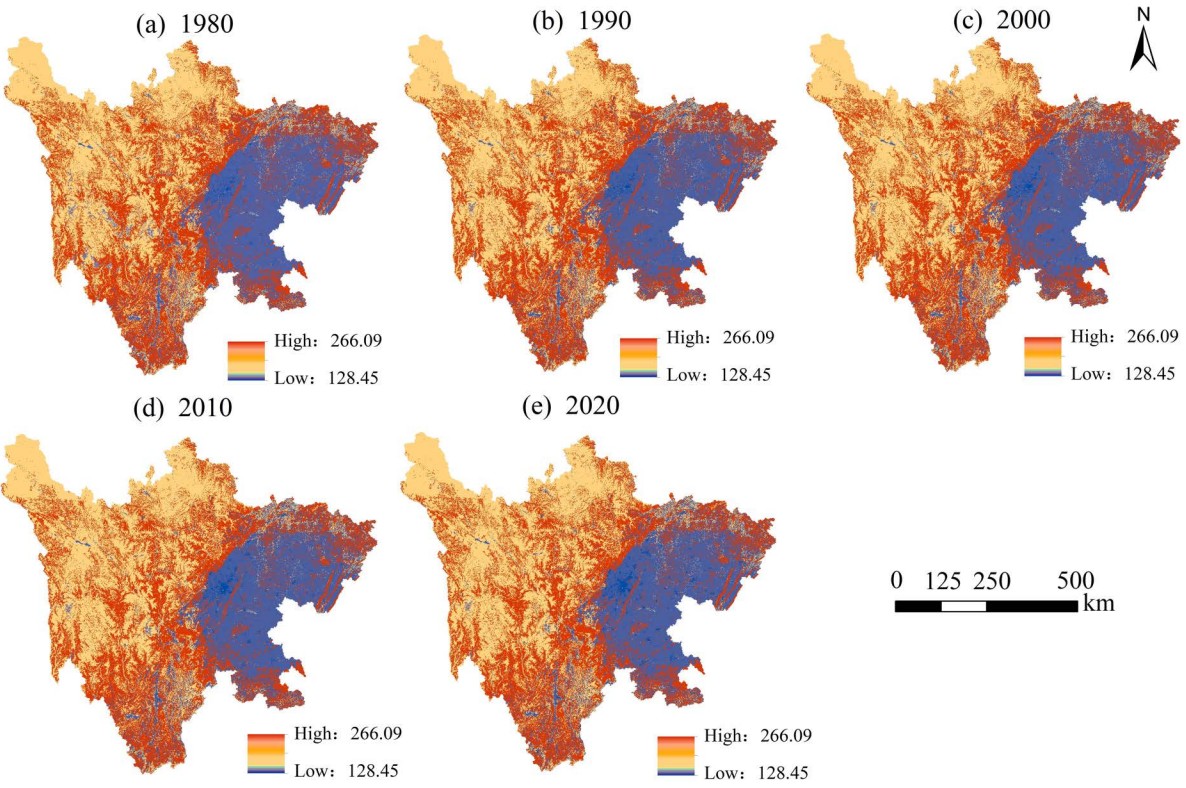

**Fig 5. Spatial distribution of carbon storage in Sichuan Province from 1980 to 2020.**

**Table 3. Carbon storage of different land use types in Sichuan Province from 1980 to 2020 (Tg).**

| Year | Land use type | | | | | | Total |
|------|----------|------------|-----------|------------|-------------------|-------------|-------|
|      | Farmland | Forest land | Grassland | Water body | Construction land | Unused land |       |
| **1980** | 1,622.17 | 4,510.44 | 2,641.31 | 96.34 | 44.16 | 287.95 | 9,202.37 |
| **1990** | 1,618.37 | 4,510.92 | 2,670.61 | 58.87 | 46.76 | 303.99 | 9,209.52 |
| **2000** | 1,612.62 | 4,492.93 | 2,679.01 | 60.09 | 53.24 | 303.64 | 9,201.53 |
| **2010** | 1,592.54 | 4,501.23 | 2,682.25 | 61.42 | 62.18 | 307.07 | 9,206.69 |
| **2020** | 1,573.41 | 4,496.79 | 2,679.09 | 66.62 | 80.63 | 306.89 | 9,203.43 |

### 4.3 Driving factors of carbon storage

**4.3.1 Contribution of land use change to carbon storage.** From 1980 to 2020, the inflow of forestland and the outflow of water bodies both contributed positively to carbon storage (Fig 7), primarily because forestland has a relatively high carbon density, whereas water bodies exhibit very low carbon density. Among all transitions, the conversion of water bodies to grassland (approximately $2.70 \times 10^{-2}$) and cropland to forestland (approximately $2.18 \times 10^{-2}$) generated the largest positive effects on carbon storage, followed by grassland to forestland ($1.64 \times 10^{-2}$) and water bodies to unused land ($1.34 \times 10^{-2}$). In contrast, the outflow of forestland exerted a negative impact on carbon storage, particularly the transition from forestland to grassland (approximately $-7.21 \times 10^{-2}$), as the carbon density of forestland substantially exceeds that of grassland. Additionally, the conversion of cropland to construction land (approximately $-1.47 \times 10^{-2}$) also resulted in declines in carbon storage.

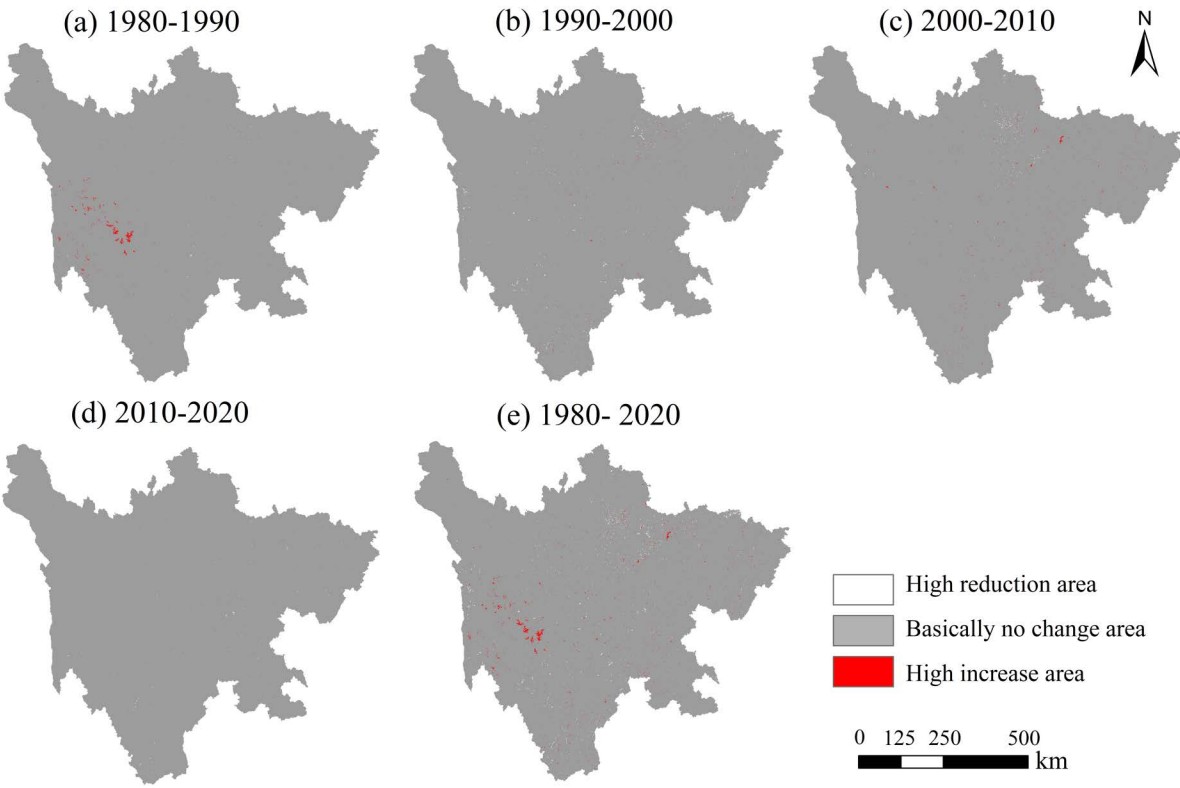

**Fig 6. Spatial variation of carbon storage in Sichuan Province from 1980 to 2020.**

**4.3.2 Spatial correlation between the LUDI and carbon storage.** This study analyzes the global and local spatial correlations between the LUDI and carbon storage at a 10 km×10 km scale to gain deeper insights into the impact of land use changes on carbon storage. After performing 999 iterations in GeoDa software, the Moran's I values for the years 1980, 1990, 2000, 2010, and 2020 were calculated as − 0.150, − 0.152, − 0.153, − 0.152, and − 0.155, respectively (Fig 8). These results indicate a consistently stable negative spatial autocorrelation between land use intensity and carbon storage over time, suggesting that higher land use intensity corresponds to lower carbon storage, and vice versa.

From 1980 to 2020, the local spatial autocorrelation distribution of carbon storage in Sichuan Province reveals a coexistence of spatial heterogeneity and spatial dependence, with this spatial pattern remaining stable over the four decades (Fig 9). High-High clusters are predominantly concentrated along the western edge of the Sichuan Basin and in the southwestern part of the study area, including locations such as Panzhihua and Xichang. Low-Low clusters mainly occur along the periphery of the study region. High-Low clusters are primarily found in the economically developed Sichuan Basin, while Low-High clusters are mainly distributed in the western areas of the study region.

**4.3.3 Impacts of other factors on carbon storage. 4.3.3.1 Analysis based on geodetector.** Factor detection results reveal significant differences in the explanatory power of various driving factors on the spatial heterogeneity of carbon storage (Fig 10). Among them, X1 (mean annual temperature) exhibits the strongest explanatory power with a q-value of 0.187, highlighting the critical influence of climate on vegetation growth and carbon cycling processes. Terrain-related factors such as X12 (DEM, q = 0.176) and X2 (slope, q = 0.152) also demonstrate prominent explanatory strength, underscoring the role of topography in shaping the spatial patterns of carbon storage. Socioeconomic and vegetation-related factors, including X6 (population, q = 0.101), X7 (NPP, q = 0.099), X11 (GDP, q = 0.093), and X8 (NDVI, q = 0.091),

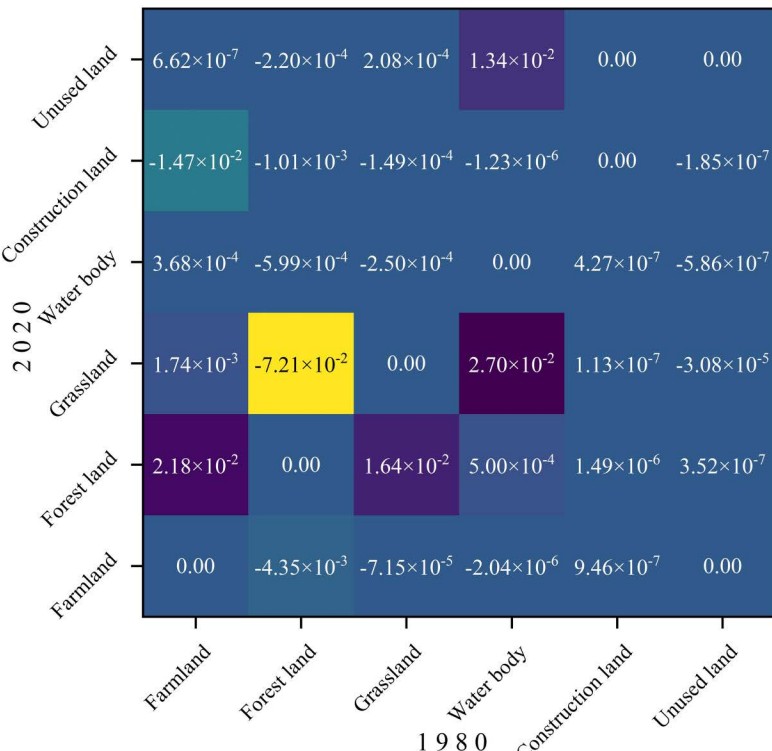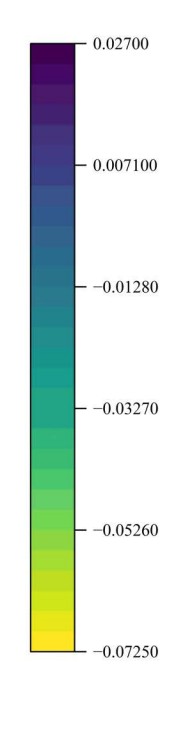

**Fig 7. Contribution of land use transitions to carbon storage in Sichuan Province from 1980 to 2020.**

show moderate explanatory power, indicating the regulatory effects of human activity intensity and vegetation productivity on carbon storage. In contrast, transportation-related factors such as X10 (distance to highways), X4 (distance to railways), X3 (distance to rivers), and X9 (distance to national highways) have relatively weak explanatory power, suggesting their limited direct impact on carbon storage. X5 (mean annual precipitation) has the lowest q-value at 0.007, reflecting its comparatively minor role in driving spatial heterogeneity of carbon storage within the study area. Overall, natural factors generally exhibit stronger explanatory power than anthropogenic factors, emphasizing the fundamental influence of soil, climate, and topography on carbon sequestration.

The interaction detector results further reveal a significant synergistic enhancement effect of multiple factors on the spatial heterogeneity of carbon storage, where interactions between different driving factors exhibit nonlinear or dual-factor amplification. This indicates that the combination of any given factor with others can strengthen its influence on carbon storage distribution (Fig 11). Notably, the interaction between X1 (mean annual temperature) and X2 (slope) yields the highest q-value (0.232), substantially exceeding their individual q-values, highlighting a coupled effect of mean annual temperature conditions and terrain slope that produces an additive impact on the spatial distribution of carbon storage. Other high q-values are observed in interactions such as X12 (DEM) with X8 (NDVI) and X12 (DEM) with X7 (NPP), underscoring the close coupling between topographic and vegetation-related factors in driving carbon storage spatial heterogeneity. Additionally, interactions between X2 (slope) and X3 (distance to rivers) at 0.161, as well as between X8 (NDVI) and X1 (mean annual temperature) at 0.218, also demonstrate strong synergistic effects, further confirming the significant roles of topography and vegetation factors in shaping carbon storage distribution.

The drivers influencing the spatial distribution of carbon storage are not simply additive in a linear manner; rather, they reflect complex interactions among multiple factors. The spatial pattern of carbon storage emerges from the intricate and

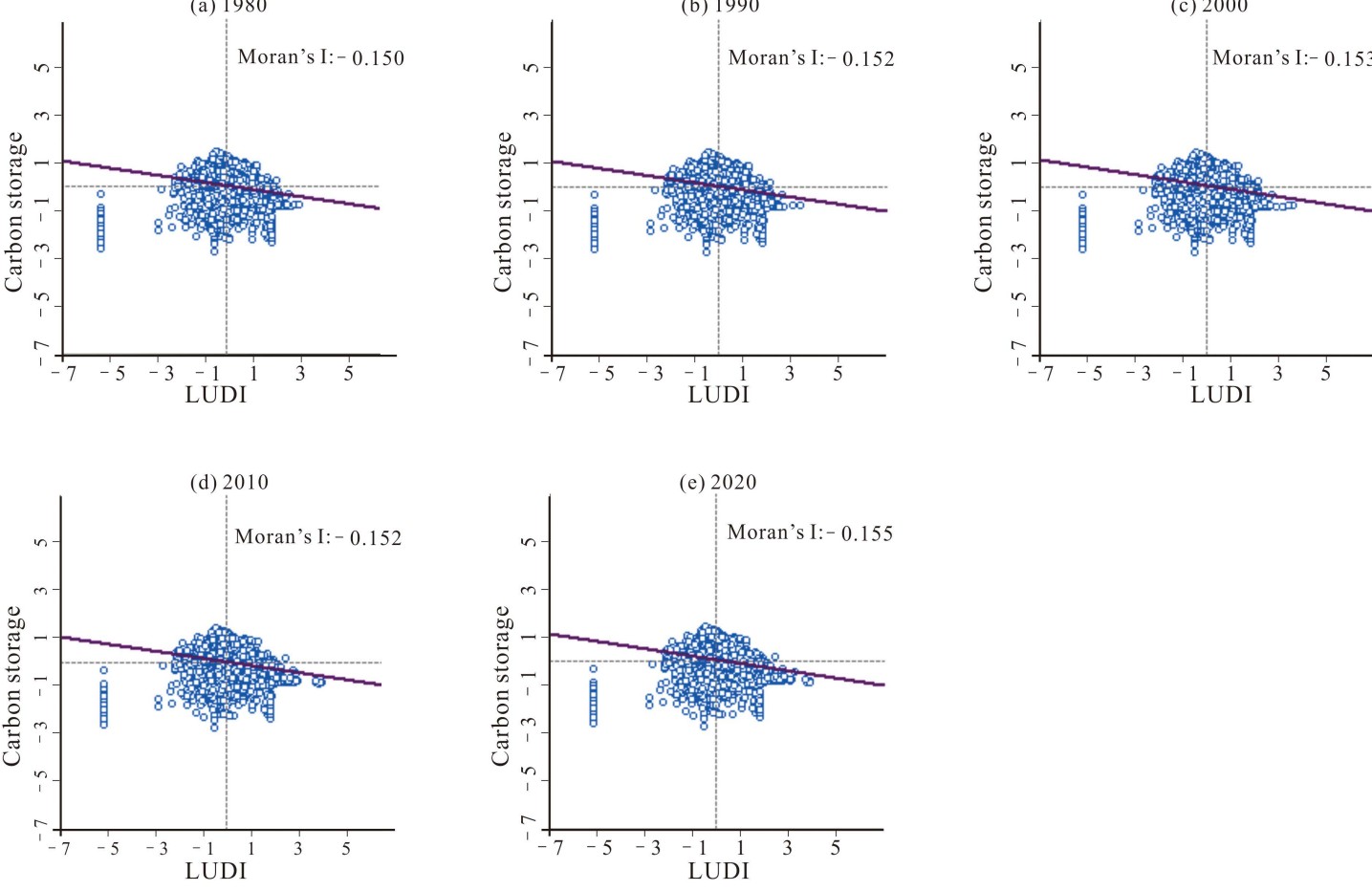

**Fig 8. Global Moran's I for carbon storage and the LUDI from 1980 to 2020.**

synergistic interplay of natural environmental elements (e.g., mean annual temperature, topography, and river) and vegetation ecological characteristics (e.g., NPP and NDVI).

**4.3.3.2 Analysis Based on Machine Learning and SHAP.** This study employed seven machine learning models to fit the relationship between twelve driving factors (X1~X12) and carbon storage. Based on accuracy and Kappa coefficient metrics (Table 4), the RF model demonstrated the best performance, and thus SHAP analysis was applied to interpret the RF model.

The SHAP analysis results (Fig 12) indicate that X1 (mean annual temperature) has the highest relative contribution ratio of 0.175, demonstrating that temperature conditions contribute most significantly to carbon storage. This is followed by X12 (DEM), with both factors exhibiting substantially greater contributions than others, highlighting the dominant influence of natural conditions on carbon storage. Factors ranked third to sixth, i.e., X8 (NDVI), X11 (GDP), X7 (NPP), and X6 (population), have contribution values ranging from approximately 0.08 to 0.10, indicating that human activities also moderately affect carbon storage. The five factors with the lowest contributions, i.e., X3 (distance to rivers), X10 (distance to highways), X4 (distance to railways), X5 (mean annual precipitation), and X9 (distance to national highways), show contributions between 0.03 and 0.06, suggesting that carbon storage is relatively insensitive to transportation factors and precipitation.

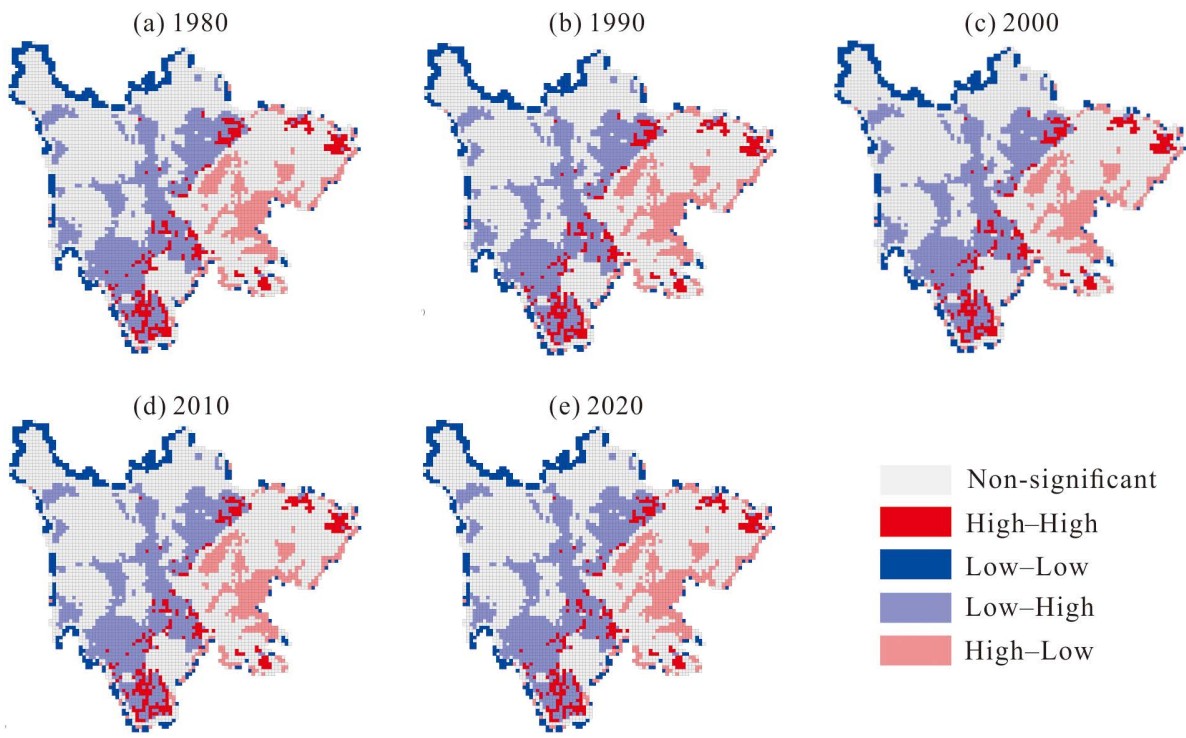

**Fig 9. LISA cluster maps of the study area from 1980 to 2020.**

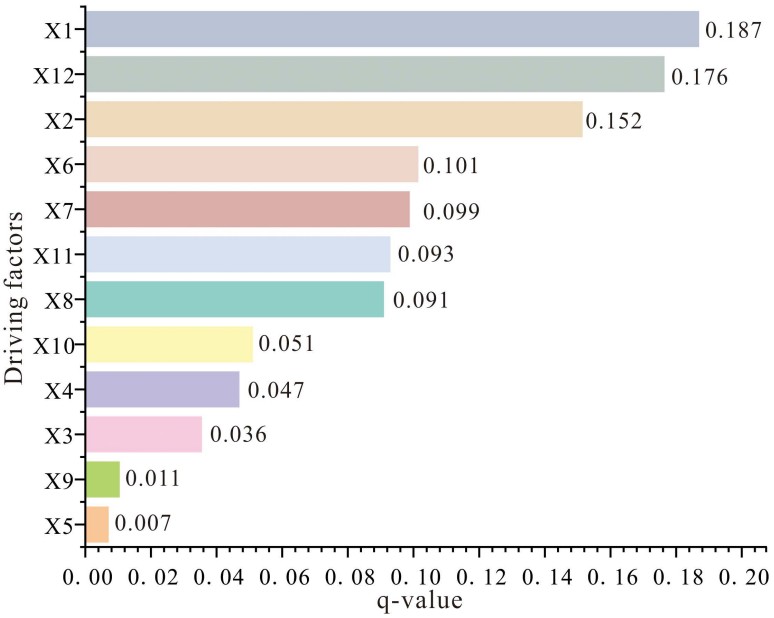

**Fig 10. Driving force of individual factors on carbon storage changes.**

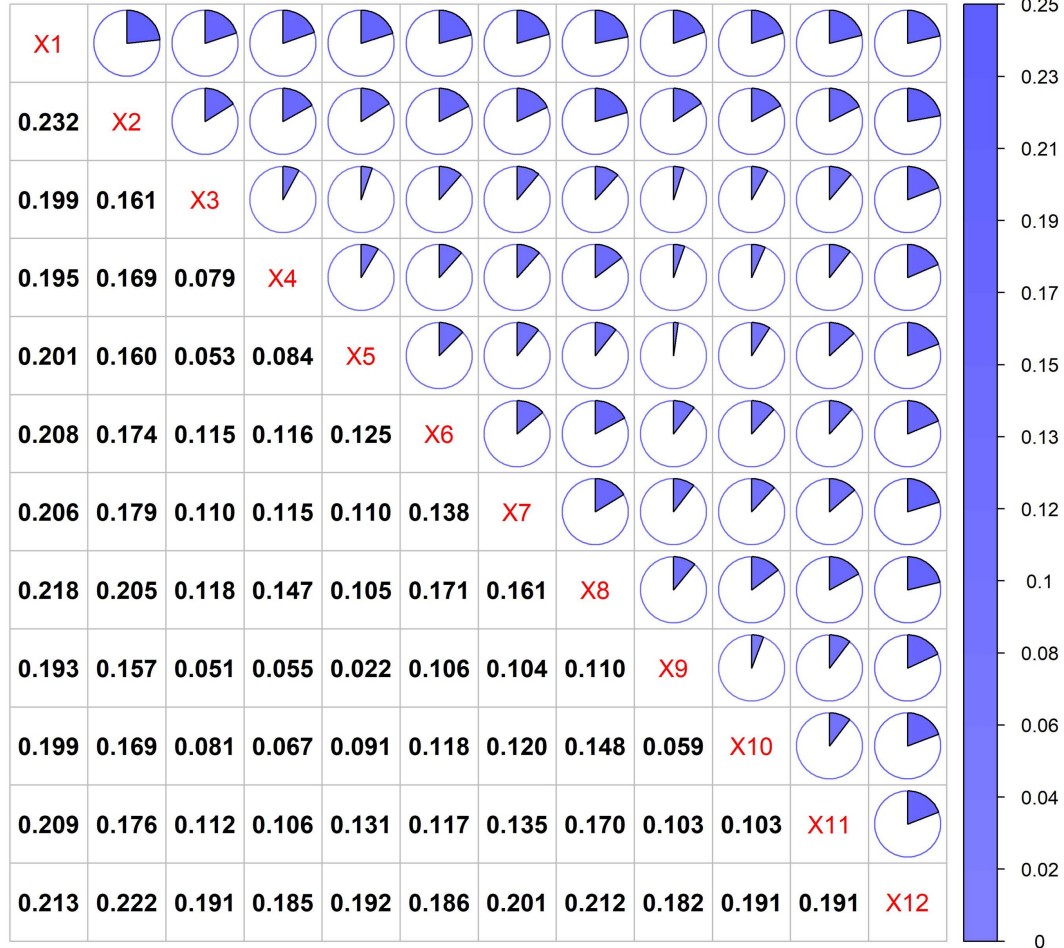

**Fig 11. Interaction detection results of driving factors.**

**Table 4. Evaluation results of seven machine learning models.**

| Evalution | Machine models | | | | | | |
|---|---|---|---|---|---|---|---|
| | LR | RF | SVM | KNN | XGBoost | CatBoost | LightGBM |
| **Accuracy** | 0.5687 | 0.8641 | 0.5581 | 0.8096 | 0.7667 | 0.7735 | 0.7462 |
| **Kappa** | 0.4824 | 0.8370 | 0.4696 | 0.7715 | 0.7201 | 0.7281 | 0.6955 |

## 4.4 Multi-scenario simulation of carbon storage in Sichuan Province from 2030 to 2050

This study conducted two separate experiments to predict land use quantity and spatial patterns for the years 2010 and 2020. Comparing the predicted land use quantities with actual data, the Markov Chain model yielded MAE values of 467 and 361 for 2010 and 2020 respectively, with an average MAE of 414. In contrast, the Linear Regression model produced MAE values of 985 and 389, averaging 687. After comparing the MAE values of the two models, this study employed the Markov chain model with the smaller MAE for land use quantity prediction.

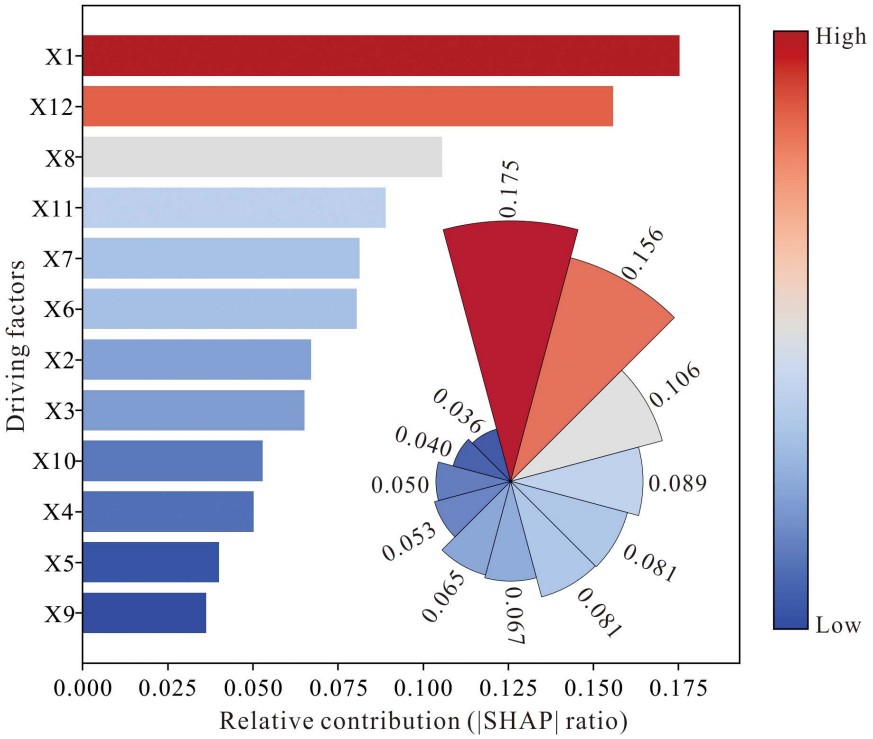

**Fig 12. Relative contributions of individual factors to carbon storage changes.**

Based on the predicted future land use quantities, spatial simulations were conducted using the PLUS and FLUS models. The PLUS model achieved Kappa coefficients of 0.9728 and 0.9634, across three simulations, with an average value of 0.9681. In comparison, the FLUS model simulations for the years 2000, 2010, and 2020 yielded Kappa coefficients of 0.9536 and 0.9432, respectively, averaging 0.9484. Given its higher accuracy, the PLUS model was selected to simulate LUCC in Sichuan Province from 2030 to 2050. Subsequently, the InVEST model was employed to assess the spatial distribution patterns of carbon storage under three future scenarios (Fig 13).

From 2030 to 2050, the spatial distribution of carbon storage in Sichuan Province continues to exhibit the characteristic "higher in the west and lower in the east" pattern (Fig 13), with significant differences in carbon storage quantities observed under different scenarios (Table 5). Under the SSP1–1.9 scenario, carbon storage is projected at 10,711.94 Tg in 2030, slightly increasing to 10,714.11 Tg in 2040 before a modest decline to 10,712.16 Tg in 2050, showing a trend of slight growth followed by a minor decrease (Table 5). In the SSP2–4.5 scenario, total carbon storage is expected to decrease steadily from 9,243.73 Tg in 2030–9,214.71 Tg in 2040 and further down to 9,202.01 Tg in 2050, reflecting a gradual weakening of ecosystem carbon sequestration capacity under moderate climate stress (Table 5). Under the SSP5–8.5 scenario, carbon storage remains relatively low and continuously declines, with values of 9,015.90 Tg in 2030, 8,989.57 Tg in 2040, and 8,980.07 Tg in 2050 (Table 5).

## 5. Discussion

### 5.1 Driving mechanism of carbon storage change

**5.1.1 Impact of land use change on carbon storage.** Between 1980 and 2020, significant changes occurred in land use types, leading to substantial variations in carbon storage. The spatial pattern of carbon storage in Sichuan Province

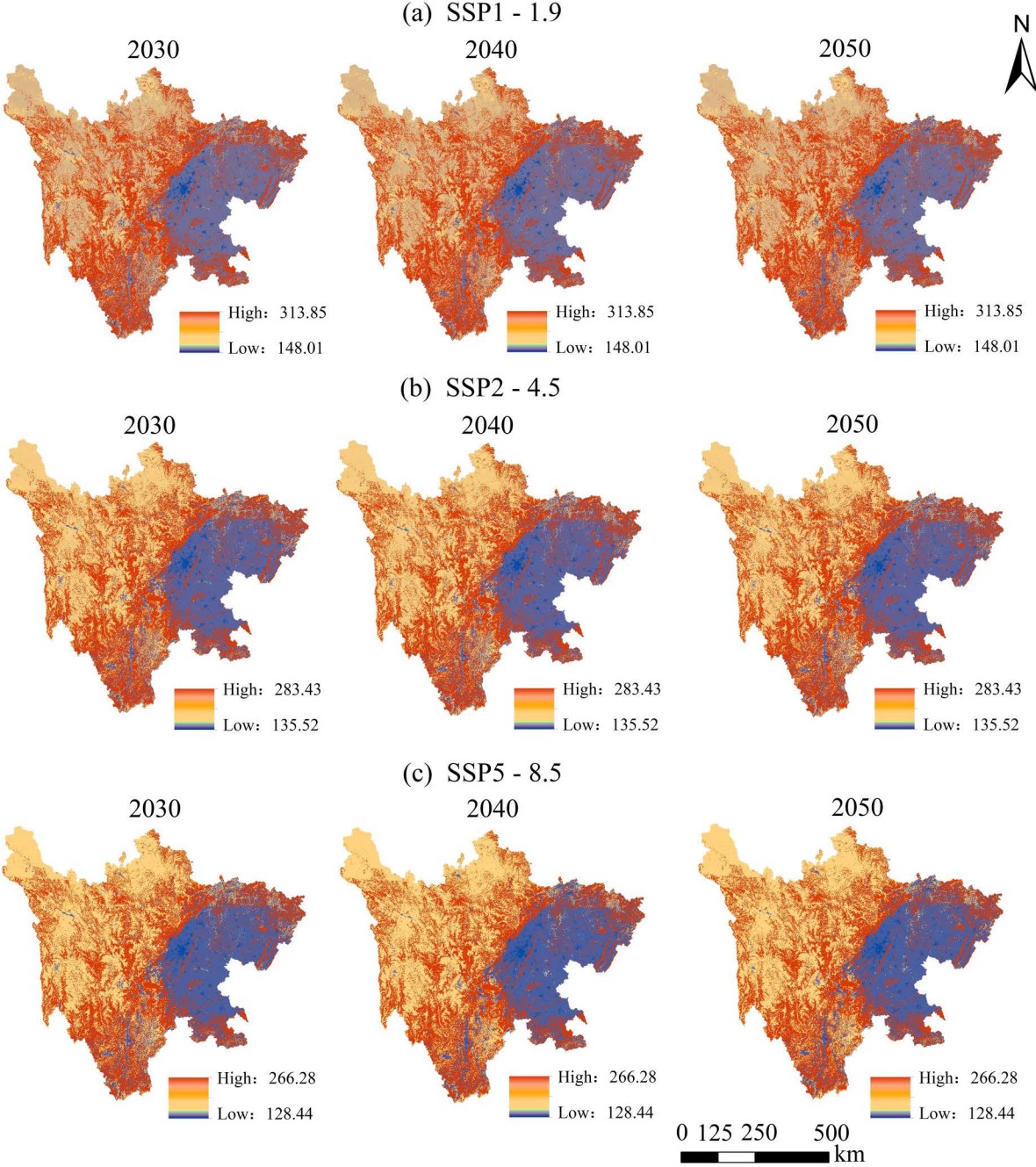

**Fig 13.  Multi-scenario simulations of carbon storage from 2030 to 2050.**

exhibits a "high in the west and low in the east" trend, attributed to the predominance of forest land and grassland in the western region, which have a stronger carbon sequestration capacity compared to the eastern areas, where farmland and construction land are more prevalent. This finding aligns closely with previous research [25,37]. Xu et al. examined carbon emissions and ecological compensation in Sichuan's main functional areas based on LUCC and identified construction land as a major carbon source, while forest land serves as the primary carbon sink [38]. The dynamic changes in forest

**Table 5. Carbon storage of all land use types under multiple future scenarios (Tg).**

| Multi-scenarios | Year | Carbon storage of different land use types | | | | | | Total |
|---|---|---|---|---|---|---|---|---|
| | | Farmland | Forest land | Grassland | Water body | Construction land | Unused land | |
| SSP1–1.9 | 2030 | 1,916.92 | 5,293.58 | 2,992.85 | 53.67 | 92.68 | 362.24 | 10,711.94 |
| | 2040 | 1,910.67 | 5,288.50 | 3,037.76 | 27.51 | 87.64 | 362.03 | 10,714.11 |
| | 2050 | 1,907.68 | 5,283.45 | 3,044.02 | 27.49 | 87.70 | 361.82 | 10,712.16 |
| SSP2–4.5 | 2030 | 1,616.96 | 4,480.50 | 2,687.31 | 49.11 | 85.07 | 324.78 | 9,243.73 |
| | 2040 | 1,610.60 | 4,475.91 | 2,684.10 | 32.38 | 87.13 | 324.59 | 9,214.71 |
| | 2050 | 1,605.57 | 4,471.35 | 2,680.88 | 31.29 | 88.51 | 324.41 | 9,202.01 |
| SSP5–8.5 | 2030 | 1,505.52 | 4,391.24 | 2,669.94 | 46.02 | 96.39 | 306.79 | 9,015.90 |
| | 2040 | 1,498.42 | 4,356.92 | 2,705.89 | 23.88 | 97.85 | 306.61 | 8,989.57 |
| | 2050 | 1,496.29 | 4,342.64 | 2,712.16 | 22.93 | 99.62 | 306.43 | 8,980.07 |

land are a key factor contributing to fluctuations in regional carbon storage, with the conversion of forest land to grassland being a major contributor to carbon loss. This conclusion is consistent with earlier studies [39–41], as the transition from high-carbon density land type (forest land) to low-carbon density land type (grassland) directly results in a decline in ecosystem carbon storage capacity. Mechanistically, the reduction in forest land, one of the land types with the highest carbon density in terrestrial ecosystems, inevitably leads to a shrinkage of the overall carbon pool, a process that is particularly pronounced in areas with intense human activity, such as eastern Sichuan [42].

The rapid expansion of construction land, with an increase of 82.6% from 2010 to 2020, has emerged as another key factor contributing to the decline in carbon storage, with the conversion of farmland to construction land being the largest contributor. This finding is consistent with the results of numerous scholars [41,43–45], who have noted that the encroachment of construction land on farmland during the urbanization process diminishes regional carbon sink functions through mechanisms such as soil disturbance and vegetation removal. Notably, this study further reveals the amplifying effect of topographical heterogeneity on this process: in the western Sichuan Plateau, transitions from water body to grassland or forest land, as well as from grassland to forest land, can result in carbon gains. In contrast, in the hilly regions of eastern Sichuan, the high intensity of human development has accelerated the conversion of forest land to farmland, leading to increased carbon release. This spatial disparity provides empirical insights for understanding the carbon cycle of ecosystems in complex topographic regions.

The results of the spatial autocorrelation analysis indicate that land use changes and carbon storage have exhibited a long-term negative spatial autocorrelation, suggesting that carbon emissions in the study area over the past 40 years are closely linked to high-intensity land use. Additionally, high-value carbon storage areas in Sichuan Province have consistently been located in the western regions, where forest land and grassland are concentrated, while low-value areas are predominantly found in the eastern regions, characterized by dense farmland and construction land. This pattern closely aligns with the spatial differentiation of land use and demonstrates a strong spatial spillover effect [37].

**5.1.2 Analysis on the driving forces of other factors on carbon storage changes.** In addition to analyzing the impact of LUCC on carbon storage, this study selected 12 factors to examine the driving mechanisms behind changes in carbon storage. In terms of explaining the driving force of a single factor on carbon storage, this paper employs both Geodetector and SHAP analysis, which enhances the credibility of the results. Both methods indicate that temperature and DEM are the most significant contributors to carbon storage. This finding is closely related to the east-west differentiation in temperature and DEM within the study area, where the eastern region features higher elevations, lower temperatures, and consequently higher carbon storage, while the western region exhibits the opposite characteristics. Moreover, the results from both GeoDetector and SHAP analyses consistently confirmed that factors such as X3 (distance to rivers), X10 (distance to highways), X4 (distance to railways), X5 (mean annual precipitation),

and X9 (distance to national highways) exhibit limited influence on carbon storage, indicating a weak relationship between carbon storage and precipitation as well as road accessibility. The remaining factors demonstrated moderate effects on carbon storage.

The analysis conducted by Tang and Peng on the factors influencing carbon storage in the Chengdu Plain, located in the central part of the study area, revealed that non-agricultural population, total agricultural output value, total service industry output value, per capita GDP, total population, and total industrial output value are the primary factors affecting carbon storage changes, with the contributions of these six factors ranging from 0.152, totaling over 0.9. In contrast, the contributions of the remaining factors, which are all natural factors, sum to less than 0.1 [46]. Xiang et al. examined the driving factors of carbon storage in the western Sichuan Plateau and found that the human activity index, mean annual temperature, and NDVI are significant contributors to changes in carbon storage; however, they only selected six factors for their analysis [22]. These two studies show considerable differences from the findings of this study, likely due to significant variations in the natural and social conditions of the study areas. Additionally, the number and type of factors selected can also influence the analysis results.

The interaction factor analysis indicates that the driving capability of dual factors on carbon storage changes is stronger than that of individual factors, such as the combinations of mean annual temperature and slope, DEM and NDVI, as well as DEM and NPP. This suggests that the distribution of carbon storage is influenced by multiple factors. Jia et al. conducted a detailed study on carbon storage in Anhui Province, China, and found that the results of interaction factor analysis exhibited nonlinear enhancement and dual-factor enhancement. They emphasized that the distribution of carbon storage is not solely influenced by a single factor but rather by the combined effects of multiple factors [47].

## 5.2 Policies and recommendations

Between 1980 and 2020, Sichuan Province experienced a reduction of 3,655 km² in farmland, with the Chengdu-Chongqing urban agglomeration, as a rapidly urbanizing area, facing particularly severe issues of farmland loss. As a fundamental element of agricultural production, the stability and quality of farmland are crucial for food security and the carbon cycle of ecosystems, necessitating the establishment of clear protection boundaries and compensation systems for farmland. In light of this, an innovative "quantity-quality-carbon sink" integrated protection model should be developed, incorporating carbon sink functions into the evaluation system for farmland protection. A compensation mechanism based on "how much is occupied, how much is cultivated" should be implemented to ensure that newly cultivated farmland meets standards for both quantity and quality while maintaining stable carbon sequestration capacity. Additionally, policies related to the delineation of "three zones and three lines" should be aligned to restrict the uncontrolled expansion of construction land under the SSP5–8.5 high-emission scenario, prioritizing the stability of ecological land such as forest land and grassland. This approach aims to mitigate the risk of carbon storage degradation through spatial planning, while also enhancing the carbon sequestration efficiency of ecosystems and controlling anthropogenic carbon emissions, taking into account the significant spatial disparities in carbon storage across Sichuan.

In terms of specific implementation recommendations, differentiated strategies should be developed for different regions and scenarios. For high-carbon storage areas such as the western Sichuan Plateau and the southwestern mountainous regions, the focus should be on maintaining forest land and grassland ecosystems, promoting native tree species with strong carbon sink capabilities, and implementing scientific logging practices to reduce carbon emissions. Additionally, ecological restoration projects should be undertaken to expand high carbon density land, thereby consolidating the region's carbon sink advantages. Furthermore, it is essential to strictly control urbanization in the eastern regions to prevent encroachment on farm and forest land, balancing economic development with carbon stock protection during land use transitions to stabilize and enhance regional ecosystem carbon storage. Lastly, a deeper understanding of the mechanisms behind localized increases in carbon storage when converting farmland to construction land is necessary,

providing a more precise scientific basis for formulating land management policies that reconcile urbanization and carbon sink protection. This approach aims to promote the sustained growth of carbon storage under the SSP1–1.9 low-emission scenario, supporting the achievement of regional carbon neutrality goals.

### 5.3 Uncertainty

Although the InVEST model provides a convenient framework for estimating carbon storage, its results are highly sensitive to the accuracy of carbon density inputs, which can directly influence the final calculations. In this study, carbon density values were derived from previous literature and further adjusted based on scenario-specific temperature and precipitation data, which may introduce uncertainties compared with actual conditions. Future research should conduct high-resolution field sampling and measurements of carbon density across the study area to improve accuracy. Moreover, this study selected 13 factors for future land-use prediction and used 12 of them to analyze the driving forces of carbon storage change; the selection of driving factors may not be fully comprehensive and inevitably involves a degree of subjectivity, potentially increasing the uncertainty of the results. Therefore, refining the criteria for selecting driving factors represents an important direction for future research. Despite these uncertainties, the policy recommendations derived from the current scenario analysis and model outputs remain scientifically valuable and practically informative. In policy formulation, on-the-ground data verification and region-specific adjustments should be incorporated to enhance applicability. Accordingly, the findings of this study still provide meaningful guidance for land-use management and carbon storage conservation in Sichuan Province and other regions with similar characteristics.

## 6. Conclusions

1. From 1980 to 2020, the land use types in Sichuan Province were primarily dominated by grassland, forest land, and farmland, with their combined area accounting for over 94%. Grassland and forest land are mainly distributed in the western part of the study area, while farmland is concentrated in the eastern region, and this spatial distribution pattern has remained largely unchanged over the past 40 years. There has been a significant reduction in the area of farmland and water body, accompanied by a notable increase in grassland and construction land. The increase in grassland area primarily results from the conversion of water body and forest land, while construction land has primarily encroached upon farmland.

2. From a spatial perspective, carbon storage in Sichuan Province exhibited a consistent "high in the west and low in the east" pattern between 1980 and 2020. The southwestern and western regions, characterized by a concentration of forest land and grassland, became high-carbon storage areas, while the central and eastern regions, with a higher proportion of farmland and construction land, displayed lower carbon storage levels. This distribution pattern has remained stable over the years. Throughout the 40-year period, the total carbon storage has remained relatively stable, fluctuating between 9,201.53 Tg and 9,209.52 Tg.

3. The conversion of water body to grassland and the transformation of farmland to forest land will both contribute to an increase in carbon storage, while the transition of forest land to grassland and farmland to construction land will lead to a reduction in carbon storage. Furthermore, higher degrees of land use intensity are associated with lower carbon storage levels. The spatial distribution of carbon storage exhibits a characteristic of coexisting heterogeneity and correlation, with High-High clusters concentrated along the western edge of the Sichuan Basin and in the southwestern part of the study area, while Low-Low clusters are primarily located around the periphery of the study area.

4. The primary driving factors behind changes in carbon storage in Sichuan Province are temperature and DEM, with accessibility and precipitation showing limited sensitivity. Human activities also influence carbon storage to a certain

extent. Interaction factors exhibit greater driving capacity for changes in carbon storage compared to individual factors, indicating that the variability in carbon storage is driven by a combination of multiple influences.

5. From 2030 to 2050, the spatial distribution of carbon storage in Sichuan Province will maintain the "high in the west and low in the east" pattern. Under the SSP1–1.9 scenario, carbon storage will reach its maximum, ranging from 10,711.94~10,712.16 Tg. This will be followed by the SSP2–4.5 scenario, with carbon storage values between 9,243.73~9,202.01 Tg. The lowest carbon storage levels will occur under the SSP5–8.5 scenario, ranging from 9,015.01~8,980.07 Tg.

## Author contributions

**Conceptualization:** Xiong Duan.

**Data curation:** Qinglian Deng, Yuqi Guan.

**Funding acquisition:** Xiong Duan, Bin Chen.

**Methodology:** Qinglian Deng, Xiong Duan.

**Software:** Qinglian Deng, Yuqi Guan, Kun Zeng.

**Writing – original draft:** Qinglian Deng, Yuqi Guan, Xiong Duan.

**Writing – review & editing:** Bin Chen, Kun Zeng.

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
