## [Decision Letter · Decision Letter 0]

12 Nov 2025

Dear Dr. Duan,

Thank you for submitting your manuscript to PLOS ONE. After careful consideration, we feel that it has merit but does not fully meet PLOS ONE’s publication criteria as it currently stands. Therefore, we invite you to submit a revised version of the manuscript that addresses the points raised during the review process.

We look forward to receiving your revised manuscript.

Kind regards,

Chenxi Li

Academic Editor

PLOS ONE

**Journal Requirements:**

1. When submitting your revision, we need you to address these additional requirements. Please ensure that your manuscript meets PLOS ONE's style requirements, including those for file naming. The PLOS ONE style templates can be found at https://journals.plos.org/plosone/s/file?id=wjVg/PLOSOne_formatting_sample_main_body.pdf and https://journals.plos.org/plosone/s/file?id=ba62/PLOSOne_formatting_sample_title_authors_affiliations.pdf 2. Please note that PLOS One has specific guidelines on code sharing for submissions in which author-generated code underpins the findings in the manuscript. In these cases, we expect all author-generated code to be made available without restrictions upon publication of the work. Please review our guidelines at https://journals.plos.org/plosone/s/materials-and-software-sharing#loc-sharing-code and ensure that your code is shared in a way that follows best practice and facilitates reproducibility and reuse. 3. Thank you for stating the following in the Acknowledgments Section of your manuscript: We are very grateful to the editors and anonymous reviewers for their comments on the manuscript. This research was funded by NSF grants (No. 42201006) of the Chinese Ministry of Science and Technology and the Natural Science Foundation of Sichuan Province (No. 2022NSFSC1177). We note that you have provided funding information that is not currently declared in your Funding Statement. However, funding information should not appear in the Acknowledgments section or other areas of your manuscript. We will only publish funding information present in the Funding Statement section of the online submission form. Please remove any funding-related text from the manuscript and let us know how you would like to update your Funding Statement. Currently, your Funding Statement reads as follows: The author(s) received no specific funding for this work.  Please include your amended statements within your cover letter; we will change the online submission form on your behalf. 4. We note that your Data Availability Statement is currently as follows: All relevant data are within the manuscript and its Supporting Information files. Please confirm at this time whether or not your submission contains all raw data required to replicate the results of your study. Authors must share the “minimal data set” for their submission. PLOS defines the minimal data set to consist of the data required to replicate all study findings reported in the article, as well as related metadata and methods (https://journals.plos.org/plosone/s/data-availability#loc-minimal-data-set-definition). For example, authors should submit the following data: - The values behind the means, standard deviations and other measures reported;- The values used to build graphs;- The points extracted from images for analysis. Authors do not need to submit their entire data set if only a portion of the data was used in the reported study. If your submission does not contain these data, please either upload them as Supporting Information files or deposit them to a stable, public repository and provide us with the relevant URLs, DOIs, or accession numbers. For a list of recommended repositories, please see https://journals.plos.org/plosone/s/recommended-repositories. If there are ethical or legal restrictions on sharing a de-identified data set, please explain them in detail (e.g., data contain potentially sensitive information, data are owned by a third-party organization, etc.) and who has imposed them (e.g., an ethics committee). Please also provide contact information for a data access committee, ethics committee, or other institutional body to which data requests may be sent. If data are owned by a third party, please indicate how others may request data access. 5. We note that Figures 1, 2, 3, 5, 6, 9 and 13 in your submission contain map images which may be copyrighted. All PLOS content is published under the Creative Commons Attribution License (CC BY 4.0), which means that the manuscript, images, and Supporting Information files will be freely available online, and any third party is permitted to access, download, copy, distribute, and use these materials in any way, even commercially, with proper attribution. For these reasons, we cannot publish previously copyrighted maps or satellite images created using proprietary data, such as Google software (Google Maps, Street View, and Earth). For more information, see our copyright guidelines: http://journals.plos.org/plosone/s/licenses-and-copyright. We require you to either present written permission from the copyright holder to publish these figures specifically under the CC BY 4.0 license, or remove the figures from your submission: a. You may seek permission from the original copyright holder of Figures 1, 2, 3, 5, 6, 9 and 13 to publish the content specifically under the CC BY 4.0 license.   We recommend that you contact the original copyright holder with the Content Permission Form (http://journals.plos.org/plosone/s/file?id=7c09/content-permission-form.pdf) and the following text:“I request permission for the open-access journal PLOS ONE to publish XXX under the Creative Commons Attribution License (CCAL) CC BY 4.0 (http://creativecommons.org/licenses/by/4.0/). Please be aware that this license allows unrestricted use and distribution, even commercially, by third parties. Please reply and provide explicit written permission to publish XXX under a CC BY license and complete the attached form.” Please upload the completed Content Permission Form or other proof of granted permissions as an "Other" file with your submission. In the figure caption of the copyrighted figure, please include the following text: “Reprinted from [ref] under a CC BY license, with permission from [name of publisher], original copyright [original copyright year].” b. If you are unable to obtain permission from the original copyright holder to publish these figures under the CC BY 4.0 license or if the copyright holder’s requirements are incompatible with the CC BY 4.0 license, please either i) remove the figure or ii) supply a replacement figure that complies with the CC BY 4.0 license. Please check copyright information on all replacement figures and update the figure caption with source information. If applicable, please specify in the figure caption text when a figure is similar but not identical to the original image and is therefore for illustrative purposes only.The following resources for replacing copyrighted map figures may be helpful: USGS National Map Viewer (public domain): http://viewer.nationalmap.gov/viewer/The Gateway to Astronaut Photography of Earth (public domain): http://eol.jsc.nasa.gov/sseop/clickmap/Maps at the CIA (public domain): https://www.cia.gov/library/publications/the-world-factbook/index.html and https://www.cia.gov/library/publications/cia-maps-publications/index.htmlNASA Earth Observatory (public domain): http://earthobservatory.nasa.gov/Landsat:
http://landsat.visibleearth.nasa.gov/USGS EROS (Earth Resources Observatory and Science (EROS) Center) (public domain): http://eros.usgs.gov/#Natural Earth (public domain): http://www.naturalearthdata.com/ 6. If the reviewer comments include a recommendation to cite specific previously published works, please review and evaluate these publications to determine whether they are relevant and should be cited. There is no requirement to cite these works unless the editor has indicated otherwise. 

Reviewers' comments:

**Comments to the Author**

1. Is the manuscript technically sound, and do the data support the conclusions?

Reviewer #1: Yes

2. Has the statistical analysis been performed appropriately and rigorously?

Reviewer #1: Yes

3. Have the authors made all data underlying the findings in their manuscript fully available?

Reviewer #1: Yes

4. Is the manuscript presented in an intelligible fashion and written in standard English?

Reviewer #1: No

**Reviewer #1:**  The research is interesting: It determined the spatiotemporal variation, driving factors and future trends of carbon storage using long-term data in Sichuan Province of Southwest China applying rigorous research approach. However I have the some comments to be considered before publication:

1.T he write-up needs improvement: make it descriptive instead of redundancy; editing is required such as citation, spacing (example line 557) ; suggestion: merge some sections, example: 3.4 and 3.5 can be merged and the heading could be ‘Driving factors of carbon storage’, and you can describe the land use change is the main factor and then the others as required (line 307, 325)

2. Full words should be there before abbreviations in the abstract (see line 31-33)

3. In the introduction, you mentioned that similar studies have been conducted in your study region, so better to well stated how noble your research methodology is, and your finding (example line 124-132)

4. Your material and method is comprehensive, but it is too long; make it concise and precise; suggestion: reduce redundancy and merge similar sections that can be separated by sentences

5. In results section, so many data but not precisely presented; should be clearly and concisely written

Line 405: Omit analysis of

Under 4.3.3, I suggest line 443-449 to be moved to material and methods section

Line 545, Section 5.1.1 improve the heading such as ‘Impact of land use change on Carbon Storage’

6. Line 622, is it land loss? I think it needs correction

7. You stated that your result is with significant uncertainty (line 651-661), so why you recommended it for policy (line 619- 650). I suggest this also needs amendment

**Do you want your identity to be public for this peer review?** For information about this choice, including consent withdrawal, please see our Privacy Policy

Reviewer #1: No

---

## [Author Response · Author response to Decision Letter 1]

16 Dec 2025

Thank you for giving us the opportunity to submit a revised version of our manuscript entitled “Carbon Storage in Sichuan Province (Southwest China) from 1980 to 2050: Spatial-Temporal Variation, Driving Factors and Future Trends ” .

We sincerely appreciate the time and effort that you and the reviewers dedicated to evaluating our manuscript. We are grateful for your valuable comments, which have substantially helped us improve the quality and clarity of the paper.

All revisions have been made accordingly and are highlighted in red in the revised manuscript. Below, we provide our point-by-point responses to each comment. If anything remains unclear or further revision is required, please do not hesitate to let us know.

Comments from the Editors and Our Responses:

Comment 1

Please ensure that the revised manuscript complies with PLOS ONE formatting requirements (including file naming)

Response:

Thank you for the reminder. We carefully reviewed the PLOS ONE formatting guidelines and revised the manuscript and file naming strictly according to the official templates to ensure full compliance.

Comment 2

Have you ensured that all custom code used in this study has been openly shared according to PLOS ONE’s code-sharing policies?

Response:

Yes. All relevant code used in this study has been deposited in a public repository and will be freely accessible upon publication (link:https://github.com/931547291-max/Qinglian-Deng).

Comment 3

Have you removed all funding information from the manuscript and prepared the correct Funding Statement for the submission system?

Response:

We have removed all funding-related information (NSF 42201006 and 2022NSFSC1177) from the manuscript, including the Acknowledgments section.

For the Funding Statement, we confirm that this study received no specific financial support. Therefore, we request the Funding Statement to read:

We apologize for any inconvenience caused by the earlier inconsistency.

Comment 4

Have you provided the minimal data set required to reproduce all study results? If not, please clarify your data-sharing approach or any restrictions.

Response:

To ensure full reproducibility of our results, we have provided the minimal data set as required. All relevant data are included in the main manuscript and its Supporting Information files. Specifically, the Supporting Information contains the following materials:

Land use/land cover maps for the years 1980–2020;

2.All driving-factor datasets used in the analysis, covering natural variables (e.g., slope, mean annual temperature, mean annual precipitation), socioeconomic factors (e.g., NPP, GDP, population), and transportation-derived indicators (e.g., distance to railways, distance to expressways, distance to national roads).

The original third-party datasets are publicly accessible through the following platforms:

Digital Elevation Model (DEM) data were obtained from the Geospatial Data Cloud (http://www.gscloud.cn), accessed on June 25, 2025.

Slope data were derived from the DEM.

Mean annual temperature, mean annual precipitation, NDVI, and soil type data were obtained from the Resource and Environmental Science Data Platform (https://www.resdc.cn), accessed on May 28, 2025.

River data were obtained from the National Catalogue Service for Geographic Information (https://www.webmap.cn), accessed on June 25, 2025.

NPP, GDP, and population data were obtained from the Resource and Environmental Science Data Platform (https://www.resdc.cn), accessed on May 28, 2025.

Railway, expressway, and national road vector datasets were obtained from the National Catalogue Service for Geographic Information (https://www.webmap.cn), accessed on June 25, 2025.

All processed datasets used in this study have also been archived in our GitHub repository (https://github.com/931547291-max/Qinglian-Deng), ensuring that both the raw and derived data required to reproduce the results are fully accessible and traceable.

Comment 5

Please confirm whether Figures 1, 2, 3, 5, 6, 9, and 13 contain copyrighted map content. If so, provide permissions or replace/remove images.

Response:

Thank you for your reminder. Figure 1 �The map in the upper-left panel shows the location of Sichuan Province within China and was generated using publicly available administrative boundary vector data.The upper-right panel was derived from the DEM, and the lower-right panel presents land-use data obtained from the Resource and Environmental Science Data Center, Chinese Academy of Sciences (http://www.resdc.cn).

Figures 2, 3, 5, 6, 9, and 13 �These figures were produced based on administrative boundaries of the study area and processed spatial datasets (e.g., land use, carbon storage). All underlying datasets were either publicly accessible or generated by the authors. No proprietary or copyrighted maps were used.

Specifically, the Digital Elevation Model (DEM) data was obtained from Geospatial Data Cloud (http://www.gscloud.cn/ Access Date: June 25, 2025), and the slope was calculated from the DEM data; data on annual average temperature, precipitation, Normalized Difference Vegetation Index (NDVI), soil type, Net Primary Productivity (NPP), Gross Domestic Product (GDP), and population were retrieved from the Resource and Environment Science and Data Center of China (https://www.resdc.cn/ Access Date: May 28, 2025); data on rivers, railways, highways, and national roads were acquired from the National Geomatics Public Service Platform (https://www.webmap.cn/ Access Date: June 25, 2025), and the distances of each element were calculated based on Euclidean distance.

All data were processed to a 1 km resolution and mapped in QGIS. Therefore, all figures are analytical results generated by the authors using open-source data, fully complying with the requirements of the CC BY 4.0 license.

https://www.tianditu.gov.cn/

https://www.webmap.cn

https://www.resdc.cn

Https://www.gscloud.cn

Comment 6

Have you reviewed the literature suggested by reviewers and evaluated their relevance?

Response:

Thank you for the reminder. We reviewed all reviewer suggestions. No additional references were required, and therefore no new citations were added.

Comment 7

Have you checked the reference list for completeness, accuracy, and possible retracted papers?

Response:

Thank you for the reminder. We thoroughly checked all references. No retracted papers were cited. All entries have been updated for accuracy and format consistency as required by the journal.

Comments from Reviewer and Our Responses

Comment 1

The write-up needs improvement: it should be more descriptive rather than redundant. Editorial revisions are also required, such as correcting citation formatting and spacing issues (e.g., line 557). In addition, it is suggested to merge some sections—for example, Sections 3.4 and 3.5 could be combined under a single heading, ‘Driving factors of carbon storage,’ where land-use change can be described as the main factor, followed by other contributing factors as needed (lines 307, 325).

Response:

Thank you for your insightful comments. We improved the writing to make it more descriptive and concise. Formatting issues such as spacing and citation style have been corrected.

We also merged Sections 3.4 and 3.5 into a new section titled “Driving Factors of Carbon Storage.” The revised section first explains land-use change as the dominant driver and then describes GeoDetector and SHAP analyses as complementary explanations.

Comment 2

In the abstract, full terms should appear before abbreviations (line 31–33).

Response:

Thank you for your suggestion. Based on your feedback, we have revised the abstract to spell out the full terms before introducing their abbreviations. Specifically, SSP1-1.9 has been revised to “Shared Socioeconomic Pathway 1-1.9 (SSP1-1.9),” SSP2-4.5 to “Shared Socioeconomic Pathway 2-4.5 (SSP2-4.5),” and SSP5-8.5 to “Shared Socioeconomic Pathway 5-8.5 (SSP5-8.5).”

Comment 3

In the introduction, you mentioned that similar studies have been conducted in your study area; therefore, it is recommended that you clearly elaborate on the novelty of your research methodology and your findings (e.g., lines 124–132)

Response:

Thank you for the reviewer’s insightful suggestions. In response, we have further clarified the methodological features and key innovations of this study in the revised manuscript. (1) More systematic spatiotemporal coverage. We constructed a province-wide LUCC and carbon storage dataset spanning 40 years (1980–2020), overcoming the common limitations of existing studies that focus mainly on local areas or short time periods. (2) A more integrated methodological framework. At the provincial scale, we coupled the Markov chain, FLUS/PLUS, and InVEST models and incorporated three CMIP6 SSP scenarios, thereby establishing an integrated “historical–current–future” simulation system. (3) More comprehensive analysis of driving mechanisms. By combining the Geographical Detector and ML–SHAP methods, we quantified the dominant natural and socio-economic drivers and their nonlinear contributions using 13 influencing factors, substantially enhancing the interpretability of the results. (4) Stronger application value. Scenario-based predictions under SSP1–1.9, SSP2–4.5, and SSP5–8.5 provide more targeted scientific support for ecological conservation and the implementation of the “dual-carbon” strategy under different development pathways in Sichuan. Overall, this study makes substantial improvements over existing research in terms of spatiotemporal scope, methodological integration, and driver analysis, offering a more systematic and replicable technical framework for carbon management in complex terrain regions.

Comment 4

Methods are comprehensive but too long; please condense and merge similar content.

Response:

Thank you for the reviewer’s constructive comments. In accordance with the suggestions, we have streamlined and consolidated the “Materials and Methods” section by removing non-essential details, merging overlapping content, and retaining only the key conditions and parameters required for reproducibility. The main revisions are as follows:

1.For the LUCC analysis and prediction section, descriptions of the land-use transition matrix, land-use degree index, Markov chain and linear regression models, and the FLUS and PLUS models have been condensed into brief definitions, core equations, and evaluation metrics. Similar explanations were merged within or across sentences to avoid repeated elaboration in different subsections.

2.For the InVEST carbon storage estimation and carbon density correction section, we retained the essential formulas, parameter sources, and scenario settings, while removing redundant descriptions.

3.For the Geographical Detector and machine learning–SHAP section, we kept the methodological framework, the main influencing factors, and the model evaluation indicators, while deleting repeated conceptual explanations.

Comment 5

Results contain too much data; rewrite more clearly and concisely. Move lines 443–449 to Methods. Improve heading in Section 5.1.1.

Response:

Thank you for your helpful suggestions. In response to the comment that “the Results section contains extensive data but lacks precision and conciseness,” we have refined and reorganized the corresponding content as follows. Highlighting key numerical results and removing secondary values. We retained only the values that have a decisive influence on the conclusions (e.g., cropland –3.0%, water bodies –30.8%, construction land +82.6%, as well as major transitions such as forest → grassland and water → grassland). Less influential categories are no longer listed individually, improving the focus of the results. (2) Reorganizing the structure based on importance. Land-use and carbon-storage changes with larger magnitudes and greater impacts are presented first, while minor changes are summarized to make the section more concise and logically coherent. (3) Strengthening the consistency between text and figures. The revised results directly reference the key values shown in the figures and tables, avoiding redundant descriptions and ensuring a clearer correspondence between textual statements and visual information.

Meanwhile, I have removed the ‘Analysis’ section at line 405, and moved lines 443–449 from Section 4.3.3 to the Materials and Methods section. In addition, I have revised the title of Subsection 5.1.1 to ‘Effects of Land-Use Change on Carbon Storage

Comment 6

Regarding line 622, does it refer to ‘land loss’? I believe this term needs to be corrected.

Response:

Thank you for your valuable comment. As you noted in line 622, the original term was not sufficiently precise. We will revise it to “farmland loss” to more accurately reflect the content of the manuscript.

Comment 7

You stated in lines 651–661 that your results involve significant uncertainty. Then why did you recommend applying them for policy-making in lines 619–650? I suggest that this section also needs to be revised

Response:

Thank you for your valuable suggestion. In the uncertainty section, we acknowledged that the results contain a certain degree of uncertainty; however, the policy recommendations we proposed are based on scenario analyses and reasonable assumptions derived from the available data, with the aim of providing useful references for decision-makers.

Nevertheless, your comment is very important. In the revision, we will clarify this point more explicitly—emphasizing that although uncertainty exists, the findings still hold referential value and can inform policy development by offering plausible scenarios. We will also note that practical implementation requires further validation and more precise data support. In future research, we will explore more effective approaches to improve accuracy.

We have made every effort to revise the manuscript thoroughly and believe the quality has been greatly improved. We sincerely appreciate the editor’s and reviewers’ valuable comments. We hope that the revised version meets your expectations.

Thank you again for your time and consideration.

Sincerely,

Xiong Duan

---

## [Decision Letter · Decision Letter 1]

22 Jan 2026

Carbon Storage in Sichuan Province (Southwest China) from 1980 to 2050: Spatial-Temporal Variation, Driving Factors and Future Trends

PONE-D-25-46315R1

Dear Dr. Duan,

We’re pleased to inform you that your manuscript has been judged scientifically suitable for publication and will be formally accepted for publication once it meets all outstanding technical requirements.

Kind regards,

Chenxi Li

Academic Editor

PLOS One

Additional Editor Comments (optional):

Reviewers' comments:

Reviewer's Responses to Questions

**Comments to the Author**

Reviewer #1: (No Response)

2. Is the manuscript technically sound, and do the data support the conclusions?

Reviewer #1: Yes

3. Has the statistical analysis been performed appropriately and rigorously?

Reviewer #1: Yes

4. Have the authors made all data underlying the findings in their manuscript fully available?

Reviewer #1: Yes

5. Is the manuscript presented in an intelligible fashion and written in standard English?

Reviewer #1: Yes

Reviewer #1: The authors addressed the comments and modified the manuscript accordingly. The only suggestion is to doublecheck the reference list and citation

**Do you want your identity to be public for this peer review?** For information about this choice, including consent withdrawal, please see our Privacy Policy

Reviewer #1: No

---

## [Editor Report · Acceptance letter]

PONE-D-25-46315R1

PLOS One

Dear Dr. Duan,

I'm pleased to inform you that your manuscript has been deemed suitable for publication in PLOS One. Congratulations! Your manuscript is now being handed over to our production team.

Kind regards,

on behalf of

Dr. Chenxi Li

Academic Editor

PLOS One